



# Nitrate and Water Isotopes as Tools to Resolve Nitrate Transit Times in a Mixed Land Use Catchment

Christina F. Radtke[1], Xiaoqiang Yang[2], Christin Müller[1], Jarno Rouhiainen[1], Ralf Merz[1], Stefanie R. Lutz[3], Paolo Benettin[4], Hong Wei[1], Kay Knöller[1, 5]

[1] Helmholtz-Centre for Environmental Research UFZ, Department of Catchment Hydrology, Germany
[2] Helmholtz-Centre for Environmental Research UFZ, Department of Aquatic Ecosystem Analysis and Management, Germany
[3] Copernicus Institute of Sustainable Development, Utrecht University, the Netherlands
[4] Laboratory of Ecohydrology ENAC/IIE/ECHO, Ecole Polytechnique Fédérale de Lausanne (EPFL), Lausanne, Switzerland
[5] Technical University of Darmstadt, Institute of Applied Geosciences, Germany

*Correspondence to*: Christina F. Radtke (christina.radtke@ufz.de)

**Abstract.** To understand the transport and fate of nitrate in catchments and its potential hazardous impact on ecosystems, knowledge about transit times (TT) and age of nitrate is needed. To add to that knowledge, we analyzed a 5-year low-frequency dataset followed by a 3-year high-frequency data set of water and nitrate isotopic signatures from a 11.5 km2 headwater

catchment with mixed land use within the Northern lowlands of the Harz mountains in Germany. For the first time, a combination of water and nitrate isotope data was used to investigate nitrate age and transport and their relation to water transit times. To do so, the numerical model tran-SAS based on Storage Age Selection (SAS) functions was extended using biogeochemical equations describing nitrate turnover processes to model nitrification and denitrification dynamics along with the age composition of discharge fluxes. The analysis revealed a temporally varying offset between nitrate and water median

transit times, with a larger offset at the beginning of wet periods due to higher proportions of young nitrate that is released more quickly with increasing discharge compared to water with larger transit times. Our findings of the varying offset between water and nitrate transit times underline the importance of analyses of solute transport and transformation in the light of projected more frequent hydrological extremes (droughts and floods) under future climate conditions.

## 1 Introduction

Due to conventional agricultural practice, the amount of nitrogen (N) applied on agricultural land very often significantly exceeds that of the actual plant uptake (Bijay-Singh & Craswell, 2021; Kirschke et al., 2019), resulting in a N surplus accumulating in soils and groundwater systems. Once organic soil nitrogen is transformed to nitrate, it is mobilized and transported by water fluxes, with the risk of contaminating receiving water bodies (Galloway et al., 2004) and fostering eutrophication in lakes and rivers that may trigger biodiversity loss. A powerful avenue to decipher nitrate dynamics are transit

time approaches, which estimate the time a water parcel or a solute has spent in a catchment since its entry via precipitation or





its forming in the soil and until it reaches the stream via discharge or leaves the catchment via evapotranspiration (Hrachowitz et al., 2016; Lutz et al., 2018; Rinaldo et al., 2015; van der Velde et al., 2010).

There is a considerable number of transit time studies that derive water age based on tracers (Benettin et al., 2020; Birkel et al., 2010; Dupas et al., 2020; Hrachowitz et al., 2016; Kleine et al., 2020; Lutz et al., 2018; Lutz et al., 2020; Nguyen et al., 35 2021; Molénat & Gascuel-Odoux, 2002; Rinaldo et al., 2015; Smith et al., 2020; van der Velde et al., 2010; van der Velde et al., 2012). Studies on nitrate transit times are, however, scarce. The relevance of water transit times in relation to nitrate transport is pointed out in the study of van der Velde et al. (2010) who investigated solute export at the Hupsel brook catchment and discovered the relationship between the dynamics of contact times of water and soil and the observed solute concentrations in stream. Nitrate transport in relation to water age was discussed by other studies (van der Velde et al., 2012; Molénat & 40 Gascuel-Odoux, 2002; Kaandorp et al., 2021; Nguyen et al., 2021; Yu et al., 2023). Nitrate removal in relation to water age was explicitly pointed out by Benettin, Fovet, & Li (2020) who estimated water age based on chloride as tracer to analyze the relationship between water age and nitrate removal. Their findings revealed that nitrate removal and water age does not correlate throughout the whole year, but an inverse relationship between nitrate removal and the release of young water was found during summer periods. They pointed out that during drier periods such as low flows during summer times, the old water 45 contribution to the stream increases. The old water contribution from deeper groundwater storages transports nitrate from a pool that underwent some extent of denitrification. Due to the latter fact, there is a negative correlation between young water fractions and nitrate removal, while nitrate removal decreases when young water fractions increase. The relation between water age and nitrate removal in riparian zones has been discussed in other studies as well (e.g. Lutz et al., 2020). However, so far, there have been few investigations of age metrics of nitrate compared to age metrics of water. None of the studies attempted 50 to estimate the explicit nitrate transit time, describing the time from its formation during nitrification in the soil until nitrate release to the stream by using isotopic signatures.

Nitrate transit times can be largely different from those of water because of chemical reactions prior to and during nitrate transport (Hrachowitz et al., 2016). The biogeochemical reactions affecting nitrate transit times are difficult to quantify, but they specifically impact oxygen isotopic signatures in nitrate, which have the potential to reveal information on nitrate transit 55 times. Nitrification of reduced inorganic nitrogen and the associated oxygen isotope exchange between reaction intermediates and ambient water were investigated in detail by Buchwald & Casciotti (2010), Casciotti et al. (2011), Boshers et al. (2019), Kool et al. (2011), Granger & Wankel (2016) and Kendall et al. (2007). Even though the nitrogen isotopic signature of nitrate is used widely to describe the sources of nitrate and the extent of biogeochemical reactions (Granger & Wankel, 2016; Kendall et al., 2007), it cannot be readily used to track nitrate age because it is governed by the highly variable isotopic signature of 60 the nitrogen compounds being transformed to nitrate such as ammonia or nitrite. In contrast, the oxygen isotopic signature of





nitrate stems from surrounding water and soil air and is incorporated into nitrate during its formation via nitrification (Boshers et al., 2019; Griffiths et al., 2016; Kendall et al., 2007; Kool et al., 2011).

With denitrification, the amount of nitrate on its flow path is reduced and by this nitrate concentrations are lowered in the stream. Denitrification along flow paths is associated with isotope fractionation resulting in a shift of both nitrogen and oxygen
isotope values in the remaining nitrate pool towards higher values with the characteristic ratio of $\delta^{18}O:\delta^{15}N$ between 0.5 and 1 (Mariotti et al., 1981; Kendall et al., 2007; Knöller et al., 2011). Hence, with longer transit times, the potential impact of denitrification on nitrate isotopes increases. While the age of water is determined by the moment when precipitation enters the soil surface, nitrification taking place in the upper soil can be considered as the initial process when the "nitrate clock" begins to tick. Naturally, there is a time lag between precipitation entering the soil and the mobilization of newly formed nitrate
following nitrification. Sebilo et al. (2013) conducted experiments that showed that the temporal offset between nitrogen input to the soil and mobilization as nitrate can take up to decades. In contrast, once it has been mobilized, nitrate can be considered to follow the same flow paths as water. However, processes like biological nitrate uptake, denitrification and mixing may have a significant influence on the median age of the nitrate pool resulting in an apparent shift between the transit times of nitrate and water.

Unravelling the time lag between N input and nitrate mobilization and transport and the differences between water and nitrate transit times is a prerequisite to better understand a catchment's capability to retain and mitigate nitrogen input for different seasons and hydrological conditions. While the time lag between input and mobilization has been addressed in previous studies (Sebilo et al., 2013; van der Velde et al., 2010), no studies have attempted to model nitrate transit times directly from the forming process until nitrate is released to the stream; hence, the offset between water and nitrate ages is largely unknown.

The objective of this study is to unravel the relationship between water age and nitrate age by hypothesizing that (i) transit times of water and nitrate have a temporal offset in a mixed land-use headwater catchment and (ii) transit times of water and nitrate are impacted by varying discharge. To test our hypotheses, we collected high frequency data (needed to investigate transit time distributions (TTD) more precisely, Stockinger et al., 2016) of isotopic signatures of water and nitrate in the Meisdorfer Sauerbach catchment. The Meisdorfer Sauerbach is part of the intensively studied terrestrial environmental
observatory TERENO (Wollschläger et al., 2017; Mueller et al., 2016; Lutz et al., 2018) and located within the Selke catchment where many studies have been conducted related to transit times of water and nitrate transport, albeit without the use of nitrate isotopes or model-based transit times (Nguyen et al., 2021; Winter et al., 2021; Yang et al., 2021; Ehrhardt et al., 2019; Lutz et al., 2020; Nguyen et al., 2022).

In addition to the conventional evaluation of the high frequency isotope data sets of water and nitrate, we adopted the transit-
time model tran-SAS v1.0 (Benettin and Bertuzzo, 2018) to simulate nitrate transit times and nitrate age by incorporating the simulation of oxygen isotope signatures and considering dominant processes like nitrification and denitrification and the





associated oxygen isotope fractionation effects. Besides high-frequency isotopic data, the model was fed with simulated hydrological and nitrate data derived from the mesoscale hydrological Model-Nitrate (mHM-N; Yang et al., 2018; Yang & Rode, 2020). We analyzed simulated water and nitrate transit times obtained by the model in order to better understand the storage and release of nitrate in mixed land-use headwater catchments. The novelty of this study is the usage of oxygen isotopic signature to simulate the forming and the degradation of nitrate at the catchment scale and by this being able to estimate the age and transit time of nitrate more accurately.

## 2 Material and Methods

### 2.1 Study area

This study was conducted in the Meisdorfer Sauerbach catchment, which is located at the north-eastern border of the Harz mountains in Central Germany (Fig. 1). The climate in the area is semi-humid with an annual precipitation of 474 mm and an annual mean temperature of 12 °C (considering the years 2013-2020). While arable land accounts for 48 % of the 11.5 km² catchment, around 46 % of the catchment are covered with forest and grassland. The remaining 6 % are urban areas (GeoBasis-DE / BKG, 2018). Dominant soil types are brown earth and podzols with a higher proportion of clay as well as luvisols with pseudogleys from loess. Alluvial soils surround the surface water bodies in the catchment (BGR, 2020). The permeability of the underlying geological sequences mainly consisting of greywacke, red sandstones and shell limestones and varies between moderate and low permeable conditions (hydraulic conductivity coefficient (kf-value) between 1E-12 and 1E-5). Aquitard sections are found throughout the catchment (BGR, 2020).




*Figure 1: The Meisdorfer Sauerbach catchment with an area about 11.5 km² is located at the north-eastern border of the Harz mountains of Germany (top left map of Germany). The red dot is the location of the autosampler for stream chemistry as well as the discharge measurement station. The orange triangles are locations of soil samples and the white circles with the black dot are locations of ground water wells. Land-use data in panel A is provided by GeoBasis-DE / BKG (2018), and elevation data in panel B in 200m resolution is taken from GeoBasis-DE / BKG (2013).*

**2.2 Sampling**

Nitrate concentrations, nitrate isotopic signatures ($\delta 15N$ and $\delta 18O$ values) and water isotopic signatures ($\delta 2H$ and $\delta 18O$ values) were measured in stream water close to the catchment outlet with varying temporal resolution. The stream water stable isotope samples were taken at fortnightly intervals from February 2017 to September 2018. Monthly samples were taken from October 2018 to April 2019. From May 2019 until March 2021, samples were generally taken as daily grab samples. During that period,



sampling was changed from a daily to a sub-daily scheme with sampling intervals between 4 and 8 hours for selected precipitation events, whenever the weather forecast predicted precipitation events in the investigation areas. Sampling schemes for nitrate concentrations and nitrate isotope signatures in stream water are in fortnightly timesteps from February 2017 to September 2018 and daily from May 2019 to March 2021. During October 2018 to April 2019, no samples were taken for the analysis of nitrate concentrations and nitrate isotopes. Besides stream water samples, composite precipitation samples were collected on a monthly base between 2013 and 2017. Fortnightly composite precipitation samples were taken from February 2017 to September 2018. Daily composite precipitation samples and sub-daily precipitation samples for selected precipitation events were collected from May 2019 to March 2021. Due to technical challenges related to the operation of the autosamplers (mainly temporary clogging of tubes and valves especially during low flow periods), only considerably reduced, variable sample volumes were collected at certain periods of times. As a consequence, a parallel analysis of nitrate isotope signatures and nitrate concentration could not be realized for all samples. In total, there are 147 measurements of nitrate isotopic signatures and 161 measurements of nitrate concentrations, while only 71 measurements of both are overlapping (from the same sampling time). Stable isotope signatures of water were analysed for 391 stream water samples and for 535 precipitation samples. In addition, seasonal groundwater (n=39) and soil moisture (n=127) samples for water isotope analysis were taken between February 2017 and September 2018 at seven different locations close to the stream in the agricultural land use section of the catchment (Fig. 1).

**2.3 Laboratory Analysis**

Water samples were filtered through a 0.45 µm filter before concentration and isotope analyses. Nitrate concentrations were measured by ion chromatography with a Dionex ICS-2000 instrument combined with an AS50 autosampler. The denitrifier method with bacteria strains of *Pseudomonas chlororaphis* was applied for determining the isotopic composition of dissolved nitrate (Sigman et al. 2001; Casciotti et al. 2002). The respective isotope measurements of the produced $N_2O$ gas were conducted with a GasBench II connected to a DELTA V Plus mass spectrometer (Thermo Scientific). The analytical precision for nitrogen and oxygen isotope measurements of nitrate were 0.4 ‰ and 0.8 ‰, respectively. International standards (USGS32, USGS34, USGS35 and IAEA NO3) were applied for correction and calibration of the raw analytical data. Stable isotope signatures of water were measured in duplicate with a liquid water isotope analyser (Picarro L2120-I). Samples were normalized to the VSMOW scale using replicate (20x) analysis of internal standards calibrated to VSMOW and Standard Light Antarctic Precipitation (SLAP) certified reference materials. The analytical uncertainty of the $\delta^{18}O$ measurement was ±0.1 ‰. Isotopic ratios are expressed in delta notation ($\delta$) relative to atmospheric nitrogen for the nitrogen isotope signature and relative to Vienna Standard Mean Ocean Water (VSMOW) for the oxygen isotope signatures of both nitrate and water:

$$\delta_{sample}[‰] = \left(\frac{R_{sample}}{R_{standard}} - 1\right) \times 1000 \tag{1}$$





With $R_{sample}$ describing the isotopic ratio of the water sample, and $R_{standard}$ describing the isotopic ratio of atmospheric nitrogen ($R_{standard}$ = 3.677x10$^{-3}$) and the VSMOW-standard ($R_{standard}$ = 2.0052x10$^{-3}$) for nitrogen and oxygen isotope measurements, respectively.

**2.4 Model set up**

To model water and nitrate transit times using a transit-time model aided by isotopic signatures, we conceptualize the catchment via a two-storage approach, i.e., an upper storage representing the upper soil layer where nitrification takes place and a routing storage representing deeper soil compartments where denitrification takes place (Fig. 2). The upper storage receives precipitation as input while water leaves as evapotranspiration and leachate to the routing storage. The routing storage

releases water as evapotranspiration or as discharge to the stream. Nitrate is formed during leaching from the upper storage to the routing storage, where denitrification takes place before nitrate is transported to the stream.

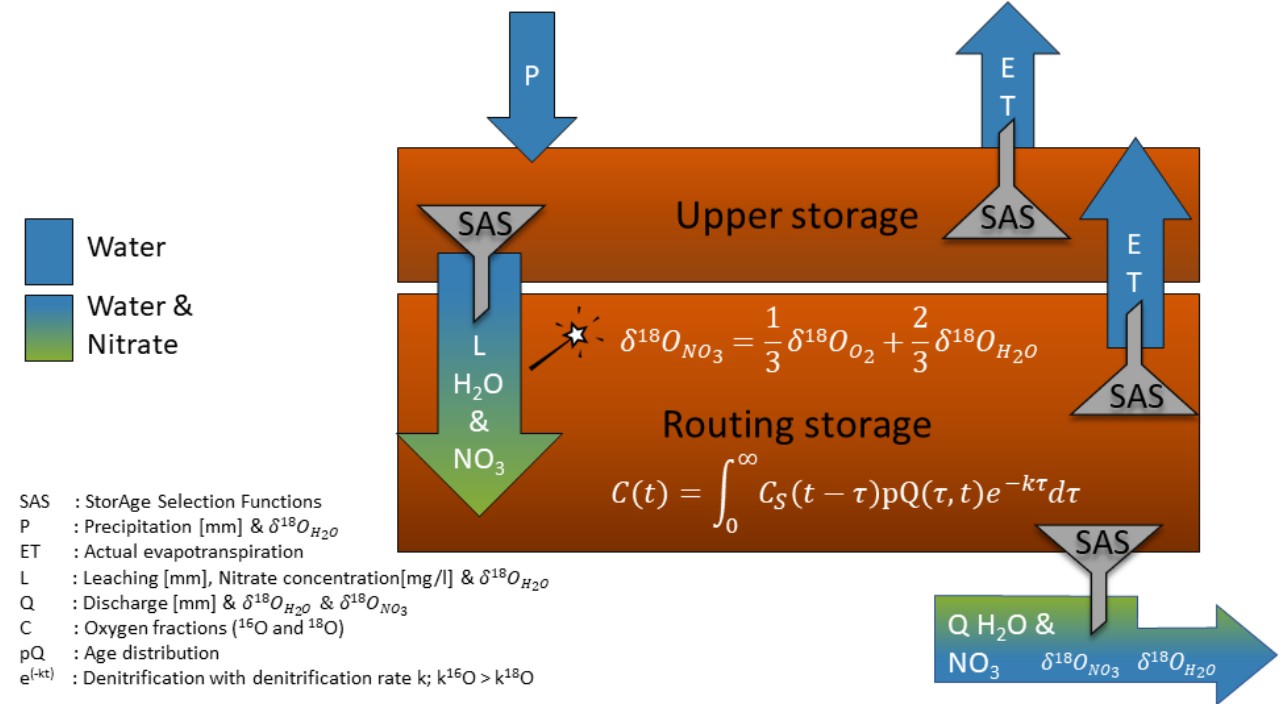

*Figure 2: Conceptual nitrate isotope model with two storages and the processes (equations) that influence the isotopic signature of $\delta^{18}O$-NO$_3$*





The water and nitrate age distributions as well as the backward transit times are modelled with the Storage Age Selection (SAS) functions approach by using a modification of the tran-SAS v1.0 model (Benettin and Bertuzzo, 2018). In the tran-SAS model, a catchment is represented as a single storage S(t) with a water-age balance that can be expressed as follows (Benettin and Bertuzzo 2018):

$$S(t) = S_0 + V(t) \tag{2}$$

$$\frac{\partial S_T(T,t)}{\partial t} + \frac{\partial S_T(T,t)}{\partial T} = P(t) - Q(t) * \Omega_Q(S_T, t) - ET(t) * \Omega_{ET}(S_T, t) \tag{3}$$

Initial condition : $S_T(T, t = 0) = S_{T_0}(t)$ \hfill (4)

Boundary condition: $S_T(0, t) = 0$ \hfill (5)

Where $S_0$ is the initial storage, $V(t)$ (mm) are the storage variations, $P(t)$ is precipitation (mm/d), $Q(t)$ is discharge (mm/d)
and $ET(t)$ is evapotranspiration (mm/d). $S_T(T,t)$ (mm) is the age-ranked storage with $S_{T_0}$ (mm) as initial age-ranked storage. The cumulative SAS functions are described as $\Omega_Q(S_T, t)$ for discharge and $\Omega_{ET}(S_T, t)$ for evapotranspiration. The (time variant and time invariant) SAS functions can be expressed as probability density functions with regard to the normalized age-ranked storage ($P_S$):

$$\omega(P_S(T,t), t) = k * (P_S(T,t))^{k-1} \tag{6}$$

$$\omega(P_S(T,t), t) = k(t) * (P_S(T,t))^{k(t)-1} \tag{7}$$

Where the catchment's water age preference for outflow is described by the parameter $k$. The catchment has a preference to
release young water if $k<1$. In case of $k>1$, the catchment tends to discharge old water. No selection preference (i.e., random sampling) is described with $k=1$. For stream water the time-variant power-law function (Eq. 7) was used in the modified nitrate isotope model. The time varying $k$ varies depending on the wetness index (Benettin and Bertuzzo, 2018). Since the focus of this study is not on the water age of evapotranspiration and due to the lack of tracer data from evapotranspiration, we applied the time invariant power law function (Eq. 6) to evaporation fluxes for the completeness of the model. By this, we got eight
parameters to be evaluated using the fit of modelled vs. observed streamflow isotope data, i.e., $kQ1$ and $kQ2$ for stream water, $kL1$ and $kL2$ for leaching water, $k\_ET1$ for evapotranspiration from the upper storage, $k\_ET2$ for the routing storage and the initial storage parameter S1.0 for the upper storage and the initial storage parameter S2.0 for the routing storage (Table 1). Two additional parameters are evaluated to represent the denitrification on nitrate isotopic signatures ($k16O$ and $\varepsilon$). More details about tran-SAS are provided by Benettin & Bertuzzo (2018).

To decouple water and nitrate age transit times, we extended the tran-SAS model by introducing the upper and lower storage, including nitrification coupled to leaching from the upper to the lower storage, and simulating denitrification in the lower storage affecting nitrate age. The model extension is based on the assumption that nitrate is formed with water leaching from



the upper storage; i.e., nitrate has the age zero when entering the routing storage. More specifically, we simulated transit time distributions for describing the age composition of discharge and nitrate export at a daily time step (Rinaldo et al. 2015). By

this, the TTD gives information about the distribution of transit times for all parcels of water and nitrate moving through the storage system. Denitrification with first order kinetics (Eq. 8) was implemented in the routing storage of the model. Denitrification is driven by microbial activity, while the extent of denitrification depends on the time that nitrate has spent in the subsurface, i.e., the TTD of water leaving the routing storage. As denitrification is a process affecting the isotopic composition of nitrate (Granger and Wankel, 2016), fractionation of oxygen isotopes is considered in the model. Hence, with

release to the stream, the nitrate isotopic signature carrying the imprint of denitrification is simulated.

To track transit times of water and nitrate, we use oxygen isotopic signatures ($\delta^{18}O$) as a tracer. During nitrification, oxygen of the ambient water is incorporated into the produced nitrate (Boshers et al., 2019; Griffiths et al., 2016; Kendall et al., 2007; Kool et al., 2011). Therefore, by tracking the transport of water isotopic signatures using $\delta^{18}O$ as tracer, it is also possible to track nitrate transit times.

The model is driven by time series of water and nitrate fluxes and isotopic signatures of precipitation, while oxygen isotope measurements from streamflow are used for calibration. Ordinary kriging of precipitation isotopic signatures with elevation was conducted in R version 4.0.5 using the package "automap" to take into account the change of isotopic signatures in precipitation with elevation. For that purpose, isotope data from other studies (Radtke et al., 2023) was used to improve the spatial distribution of isotopic signatures. The daily isotopic signature was extracted as spatial mean over the catchment area.

Hydrological fluxes (i.e. evapotranspiration, discharge and leaching from the upper storage) and associated nitrate concentrations (i.e. nitrate in leaching flux) were computed using the spatially explicit distributed mHM-Nitrate model (mesoscale Hydrological Model for Nitrate; Yang & Rode, 2020), a fully distributed nitrate transport and removal model. A detailed description of the model structure, conceptualization and calibration of mHM-Nitrate, which combines concepts of the hydrological model (mHM) (Samaniego et al., 2010) and the Hydrological Predictions for the Environment (HYPE) model

(Lindström et al., 2010), is provided by Yang & Rode (2020). Yang & Rode (2020) applied mHM-Nitrate at daily time step to the larger Selke catchment, of which the Meisdorfer Sauerbach is a tributary, using precipitation and temperature patterns interpolated from observed data (Rauthe et al., 2013; DWD, 2017), and calibrated the model against observed discharge as well as in-stream nitrate concentrations at three main stem stations of the Selke catchment (Yang et al. 2018).

**2.5 Nitrate isotope model**

To get information on nitrate ages, the tran-SAS model is extended to account for nitrification and denitrification. The nitrate isotope model is based on oxygen isotopic signatures of nitrate ($\delta^{18}O$-$NO_3$) instead of the nitrogen isotopic signature ($\delta^{15}N$-$NO_3$). This is due to the fact that the initial oxygen isotope signature of nitrate is determined by the water oxygen isotope value during the formation of nitrate (Boshers et al., 2019; Buchwald and Casciotti, 2010; Granger and Wankel, 2016; Griffiths et al., 2016; Kool et al., 2011). Hence, we assume it can be utilized as a time stamp for nitrate origin. In contrast, the initial $\delta^{15}N$-





NO₃ is mainly driven by the nitrogen isotope signatures of the precursor nitrogen-bearing substrates (inorganic reduced nitrogen, organic nitrogen) that are processed during nitrification, the main formation process of nitrate. Therefore, nitrogen isotope signatures in nitrate are not suitable for revealing age information of dissolved nitrate.

The isotopic signature of NO₃ formed during nitrification can be described by Eq. (8) (Kool et al., 2011; Boshers et al., 2019):

$$\delta^{18}O_{NO_3} = \frac{1}{3}\delta^{18}O_{O_2} + \frac{2}{3}\delta^{18}O_{H_2O} \tag{8}$$

with $\delta^{18}O_{O_2}$ being the isotopic signature of soil air (parameter between 22 and 29 ‰ according to Mayer et al. (2001)) and

$\delta^{18}O_{H_2O}$ being the isotopic signatures of leaching water. The oxygen isotopic signature of soil air is often determined with a value of 23.5 ‰ (Boshers et al., 2019; Griffiths et al., 2016; Kendall et al., 2007, Kool et al., 2011), although the isotopic signature of soil air can vary due to different influences. For example, Kendall et al., (2007) mentioned that the isotopic signature of soil air can be lower due to photosynthesis or higher due to respiration by microbes which is in line with findings by Mayer et al., (2001), who observed variations of isotopic

signature of soil air between 22 and 29 ‰. Moreover, initial oxygen nitrate isotope signatures fixed during nitrification may undergo an alteration that could bias the extracted age information. This alteration is related to an isotope fractionation during denitrification or a secondary oxygen isotope exchange of process intermediates (NOₓ) with the ambient water (Granger and Wankel, 2016). While the potential impact of denitrification is considered in the model, a secondary exchange of oxygen isotopes is not taken into account because of the high uncertainty related to the reliable numerical prediction of that exchange

in combination with multiple environmental and ecological parameters driving the exchange process. Nevertheless, we expect the alteration of oxygen isotope signatures by secondary isotope exchange to have a minor impact on the nitrate age simulations. Therefore, for the sake of simplification, we neglected that process in our model and, for reasons of simplification, set the isotopic signature of soil air to 23.5 ‰.

The extent of denitrification, described with first order kinetics, was determined by the TTDs of the water transporting nitrate

through the routing storage. We simulated transport and denitrification for the two isotopic species of nitrate (¹⁶O and ¹⁸O) separately. This leads to the following expression for nitrate isotope values in the stream:

$$C(t) = \int_0^\infty C_S(t - \tau)pQ(\tau, t)e^{-k\tau}d\tau \tag{9}$$

where $C(t)$ is the respective concentration of the two different oxygen isotopes of nitrate (¹⁸O, ¹⁶O) at the catchment outlet, $C_S(t - \tau)$ describes the respective concentrations of ¹⁸O and ¹⁶O of nitrate in the routing storage per timestep, and $pQ(\tau, t)$ represents the transit time distribution of the water transporting nitrate through the routing storage at time *t*. The first order

kinetics expression $e^{-k\tau}$ describes denitrification during transport through the routing storage, where $k$ is the degradation rate constant and $\tau$ describes the transit time. The resulting oxygen isotope signature in the stream was calculated using Eq. 1. The nitrate concentration obtained by the nitrate isotope model is calculated by the concentration sum of the two isotopic species:



$$C(t)_{NO3} = C(t)_{18O} + C(t)_{16O} \tag{10}$$

with the respective concentrations of the two different oxygen isotopes of nitrate ([18]O, [16]O) at the catchment outlet for each time step and $C(t)_{NO3}$ as resulting nitrate concentration at time step t in mg/l.

Isotope fractionation during denitrification with rate constant $k$ (see Eq. 9) is determined by the isotope fractionation factor ($\alpha$):

$$\alpha = \frac{k_{18O}}{k_{16O}} \tag{11}$$

The corresponding range of values for the fractionation factor and degradation rate constant of the light oxygen isotope species ($k_{16O}$) were taken from Granger & Wankel (2016). The range of fractionation factors was given as epsilon values ($\mathcal{E} \approx$ 1000ln($\alpha$) which were between -5‰ and 5 ‰, while degradation rate constants ($k_{16O}$) were between 0.0 and 0.9 [day[-1]]. We

only considered isotope fractionation for oxygen isotopes in nitrate.

**2.6 Model evaluation**

The statistical method generalized likelihood uncertainty estimation (GLUE) (Beven and Binley, 1992; Beven and Binley 2014) was applied to evaluate the model performance of randomly selected 10.000 parameter sets of the catchment. The GLUE approach is a valuable tool to evaluate the parameter uncertainty and equifinality. Multiple sets of parameter values can be

evaluated within the limitations that are given for a model (Beven and Binley, 1992; Beven and Binley 2014). In this study, we selected the 10% best simulations (Table 1) considering the Kling-Gupta-Efficiency (Gupta, Kling et al. 2009) between observed and simulated isotopic signatures in the stream (Eq. 12).

$$KGE = 1 - \sqrt{(r-1)^2 + \left(\frac{\sigma_{sim}}{\sigma_{obs}} - 1\right)^2 + \left(\frac{\mu_{sim}}{\mu_{obs}} - 1\right)^2} \tag{12}$$

where $\sigma_{obs/sim}$ is the standard deviation in observations/simulations, $\mu_{obs/sim}$ is the mean of all observations/simulations and $r$ is the Pearson correlation coefficient between observations and simulations. KGE=1 indicates perfect agreement between

simulations and observations. A spin up period repeating the period 2013 to 2016 three times is used to minimize the effect of initial conditions. The initial condition of the isotopic signature $\delta^{18}$O is set to -8 ‰ in both storages.

The mean simulation of all 10% best parameter combinations according to the highest KGE for water isotopic signatures in stream were selected for further analysis.





Table 1: Parameter ranges of the 10% best simulations according to the KGE of the water isotopic signatures in stream (amount
simulations = 212)

| Parameter | Parameter description | Lower boundary | Upper boundary |
|---|---|---|---|
| kL1 [-] | Time-variant power-law function parameter for leaching water | 0.01 | 0.09 |
| kL2 [-] | Time-variant power-law function parameter for leaching water | 0.01 | 0.10 |
| k_ET1 [-] | Time-invariant power-law function parameter for evapotranspiration, upper storage | 0.60 | 0.99 |
| S1.0 [mm] | Initial condition of the water volume in the upper storage | 80 | 190 |
| kQ1 [-] | Time-variant power-law function parameter for discharge | 0.10 | 0.47 |
| kQ2 [-] | Time-variant power-law function parameter for discharge | 0.10 | 0.21 |
| k_ET2 [-] | Time-invariant power-law function parameter for evapotranspiration, routing storage | 0.60 | 0.99 |
| S2.0 [mm] | Initial condition of the water volume in the routing storage | 153 | 499 |
| $k_{16_O}$ | Denitrification rate of the light oxygen isotope [day$^{-1}$] | 0.01 | 0.10 |
| ε | Epsilon as fractionation factor for the isotopic signature [‰] | -5 | 5 |





## 2.7 Evaluation of isotopic enrichment

To evaluate the occurrence of microbial activity such as denitrification, the Rayleigh plot can be used. Isotopic enrichment during denitrification can be expressed by a simplified Rayleigh equation (Eq. 13):

$$\delta^{15}N_t = \varepsilon \, ln \left( \frac{C_t}{C_{max}} \right) + \delta^{15}N_{C-max} \tag{13}$$

where $\varepsilon$ is the isotopic enrichment factor, $t$ expresses the nitrogen isotopic composition and nitrate concentration at any time step and $max$ refers to the highest measured concentration and the isotopic composition determined at that particular timestep.

## 3 Results and discussion

### 3.1 Evaluation of measured isotope data

The isotopic signature of $\delta^{18}O$ of water in stream shows a damped signal compared to the isotopic signature of water in precipitation. Only during event sampling with higher sampling frequencies, higher fluctuations of isotopic signatures are visible in stream water. The isotopic signature in precipitation varies between -21.3 ‰ and 3.4 ‰ for $\delta^{18}O$ while the isotopic signature in stream water varies between -11.2 ‰ and -6.7 ‰ for $\delta^{18}O$. Soil moisture and groundwater samples for $\square^{18}O\text{-}H_2O$
varied from -14.9 ‰ to -1.0 ‰ and from -9.2 ‰ to -7.9 ‰, respectively. For instream nitrate, $\delta^{18}O\text{-}NO_3$ varied between -3.3 ‰ and 33.1 ‰ and $\delta^{15}N\text{-}NO_3$ varied between 3.5 ‰ and 27.4 ‰ (Fig. 3). Measured $NO_3\text{-}N$ concentrations in stream water varied between 0.25 mg/l and 9.6 mg/l.





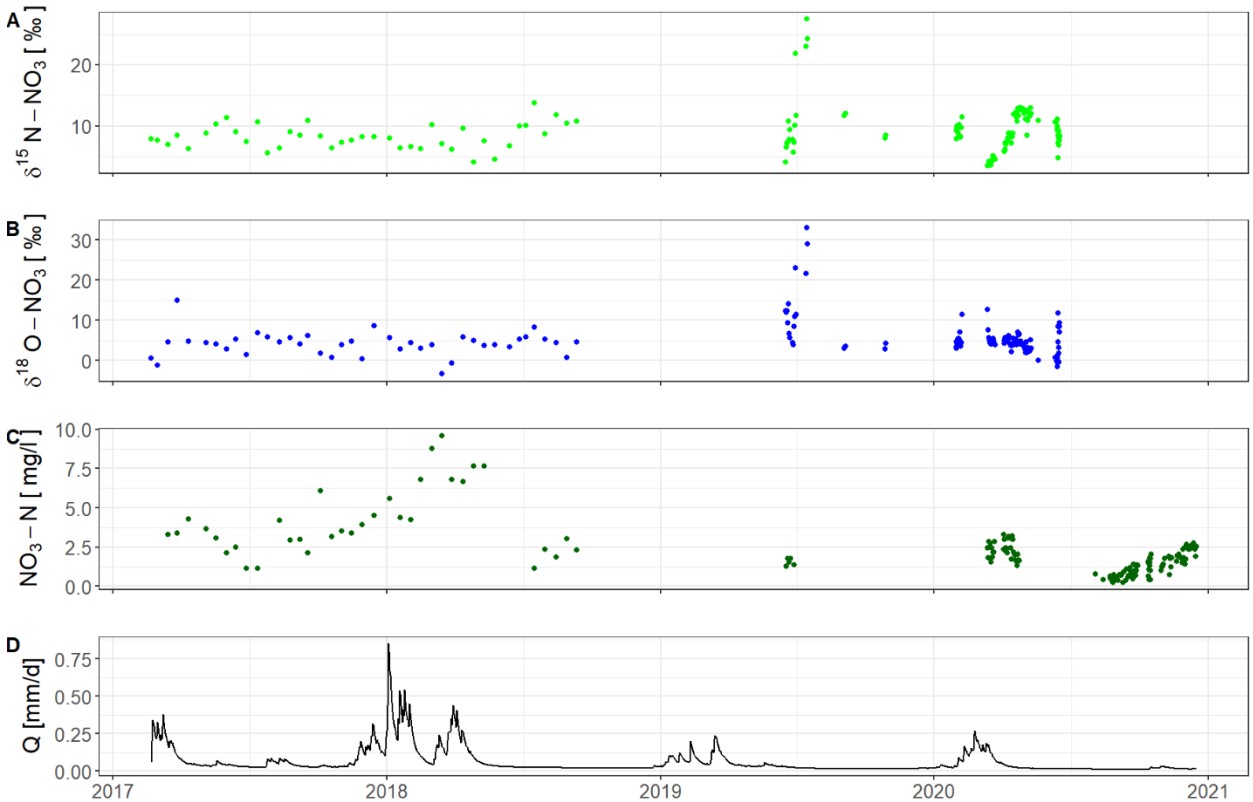

Figure 3: Isotopic signatures of nitrate (A, B) as well as the concentration of NO$_3$-N in stream water (C) and the discharge (Q)
[mm/d] (D) for the observation period

Isotopic signatures of nitrate that were measured fortnightly show less strong variations compared to the isotopic signatures of
nitrate that have been measured sub-daily to daily (Fig. 3). In 2019, the highest isotopic signatures of nitrate can be found
during the daily sampling scheme. Considering the NO$_3$-N concentrations, the pattern is different. During the fortnightly
sampling campaign, NO$_3$-N concentrations rose up to 10 mg/l while during the sub-daily to daily sampling scheme, NO$_3$-N
concentrations rose up to 3.5 mg/l. In general, NO$_3$-N concentrations during the high-frequency sampling scheme were lower
compared to the low-frequency sampling scheme. With a delay, the NO$_3$-N concentrations increase in response to an increasing
discharge, which can be seen in the beginning of 2018 and 2020. The isotopic signatures show a minor change in their
composition in response to changing hydrologic conditions: the oxygen isotopes show a more scattered pattern during high
flows, while nitrogen isotopes show a decrease in their isotopic signatures during high flows, which can be seen both in the
beginning of 2018 and 2020. These findings indicate that nitrate is flushed out of the storage system due to high flow events.
Storage selection functions that are used in numerical models such as the tran-SAS (Benettin and Bertuzzo, 2018) can reflect
these varying conditions; therefore, we decided to use the SAS approach for further investigation of nitrate and its age derived
from                                          nitrate                                          isotopic                                          compositions.





Since nitrate can derive from different nitrogen sources, it is not straightforward to determine the exact nitrogen source. In Fig.

4, the possible sources of nitrate in the Meisdorfer Sauerbach catchment according to its isotopic values are shown.

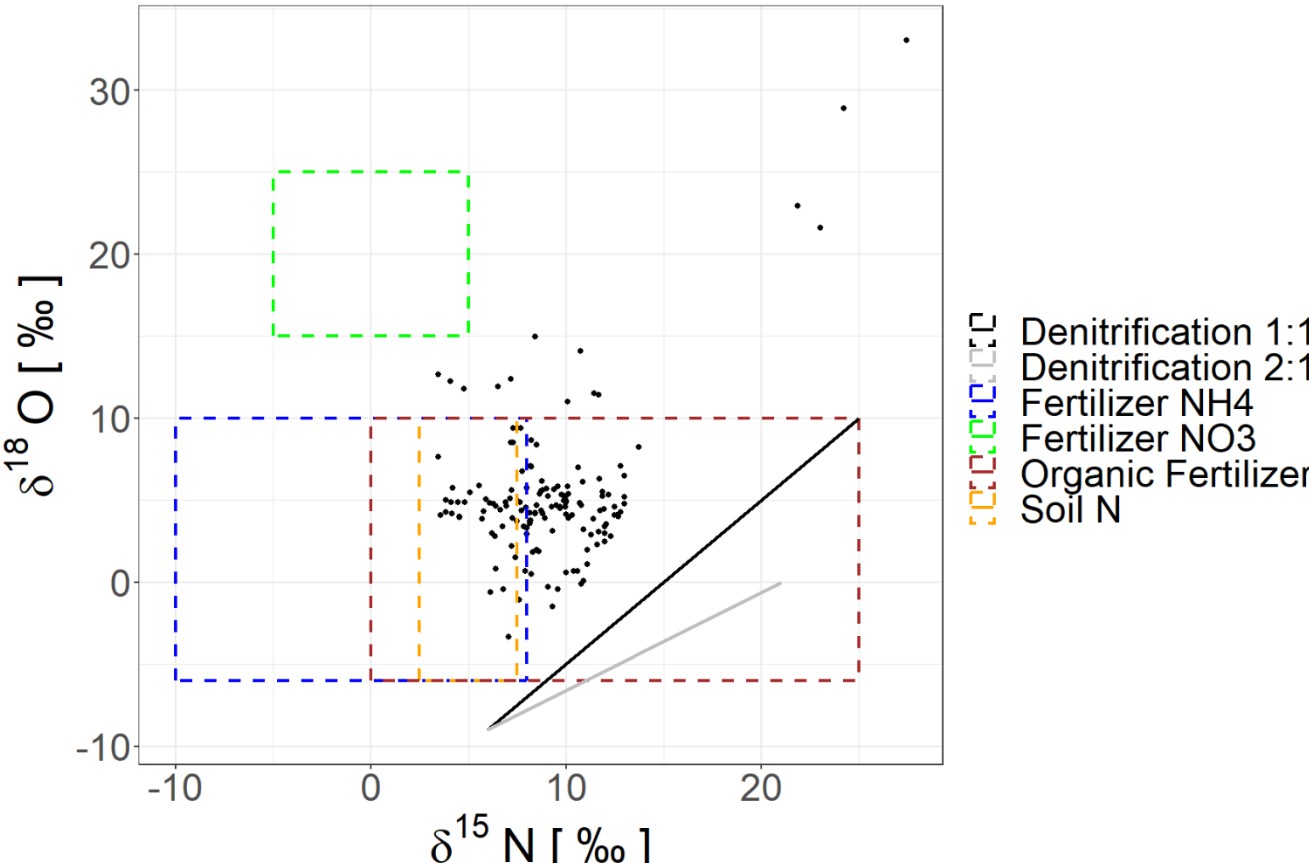

Figure 4: Isotopic signatures of nitrate in stream water in the Meisdorfer Sauerbach catchment, indicating different possible nitrate sources.

In the Meisdorfer Sauerbach catchment, the relation between both isotopic signatures ($\delta^{15}$N and $\delta^{18}$O of nitrate) show possible sources mainly from fertilizer such as organic fertilizer and ammonium fertilizer as well as ammonium formed in the soil zone (Fig. 4), with boundaries of the isotopic signatures of the sources taken from Kendall et al. (2007) and adapted to the precipitation isotopic signature that have been measured in the catchment. There is no clear sign of denitrification evident in the isotopic measurements. Nevertheless, denitrification cannot be excluded, since some gaps in the measured time series due

to a lack of sample water are present (Fig. 3). To further investigate whether the dataset points towards the occurrence of denitrification, we compare monthly distributions of isotopic signatures of nitrate with the monthly distributions of measured nitrate concentrations (Fig. 5).





Figure 5: Monthly distribution of the isotopic signature (panel A) $\delta^{18}O$-$NO_3$, (panel B) $\delta^{15}N$-$NO_3$ and (panel C) nitrate-N
concentration during the years 2017-2020





The monthly distribution of $\delta^{18}$O-NO$_3$ in comparison to the nitrate-N concentration shows increasing isotopic signatures for decreasing nitrate-N concentrations (Fig. 5). Even though this relationship is not strong (Pearson correlation: -0.12), the behavior is visible for instance in 2019. The highest variability of oxygen isotopic signatures was found in the summer of 2019, where low nitrate-N concentrations under 10 mg/l were observed. In the beginning of 2018, the nitrate-N concentrations

increased, and the most depleted oxygen isotopic signatures coincide with the highest nitrate-N concentrations. The observed pattern points towards different mechanisms of nitrate formation depending on hydrologic conditions. After fertilizer application, microorganisms in the soil zone consume the infiltrated substances of the fertilizer application and nitrify them to nitrate. The isotopic signature of the nitrified nitrate, which can be taken up by plants as well as flushed out of the system due to high flow periods, is dominated by the soil oxygen isotope signature and therefore shows more decreased oxygen isotopic

signatures. In summer, with higher temperatures, nitrate consuming processes such as denitrification occur under moist conditions (Kendall et al., 2007; Kaneko and Poulson, 2013; Granger and Wankel, 2016). Depleted nitrate-N concentrations during summer months with increasing oxygen isotopic signatures are indicators for denitrification that occurred in the soil zone under high temperatures and high soil moisture. The measured samples in the Meisdorfer Sauerbach show only short-term periods; therefore, a continuous time series analysis is not possible. Considering the samples we have, some indications

(variations of isotopic signatures; fractionation) of microbial processes can be observed such as nitrification and denitrification, even though these processes likely occur irregularly and cannot be confirmed by the dual isotope plot or the relation between nitrate concentrations and isotopic signatures of $\delta^{15}$N.

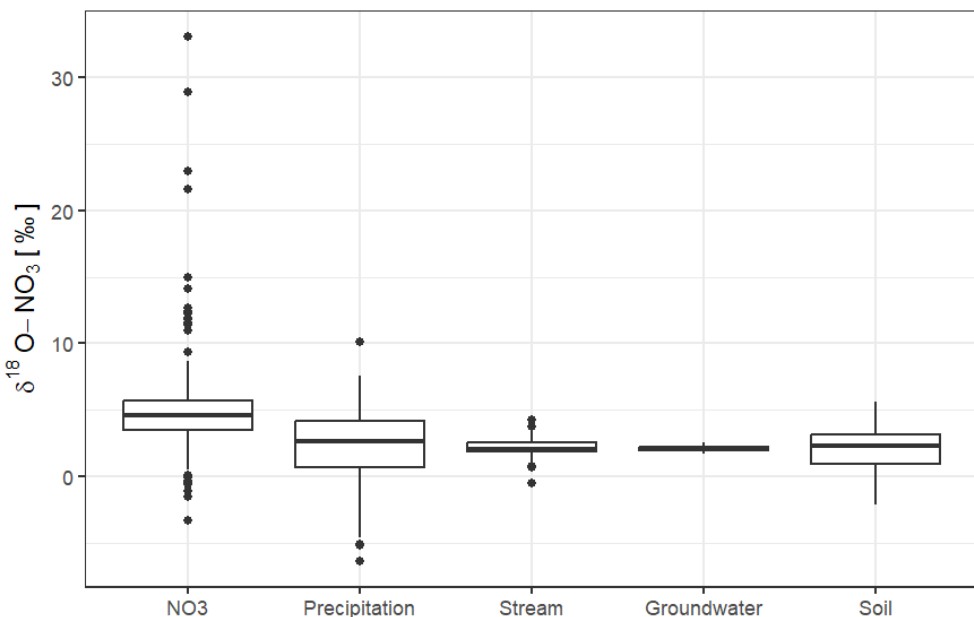

*Figure 6: Comparison of the measured range of $\delta^{18}$O-NO$_3$ in stream water with calculated ranges of $\delta^{18}$O-NO$_3$ based on the*
*$\delta^{18}$O-H$_2$O ranges observed in different compartments using Eq. (8)*





Using Eq. (8) with the isotopic signature of 23.5 ‰ for $\delta^{18}O\text{-}O_2$ and the measured $\delta^{18}O\text{-}H_2O$ values of precipitation, groundwater, soil moisture and stream water results in hypothetical $\delta^{18}O\text{-}NO_3$ values that reflect the compartment in which nitrification has occurred. Interestingly, Fig. 6 illustrates that the observed range of $\delta^{18}O\text{-}NO_3$ is larger than the calculated ranges for the different compartments. Moreover, while the calculated means are very close to each other (2-3 ‰), the mean

of the observed oxygen isotopic range in nitrate is significantly higher (4.5 ‰). The most obvious reason for the differences between calculated and observed ranges is a secondary biogeochemical impact on the oxygen isotopic composition of the nitrate pool that to some extent disturbs the original isotopic signature fixed during nitrification. A negative deviation from the calculated range could be associated with oxygen isotope exchange of reaction intermediates (primarily nitrite) with the ambient water. This scenario is especially likely for temporally or locally variable biogeochemical conditions favouring both

the reductive pathway of nitrate reduction and the oxidative pathway of nitrite oxidation (Granger and Wankel, 2016). A positive deviation of observed $\delta^{18}O\text{-}NO_3$ from computed $\delta^{18}O\text{-}NO_3$ is most likely caused by the impact of denitrification. The overall impact of denitrification on the catchment scale can be evaluated by integrated data analysis as shown below (Kendall et al., 2007). Considering all measured isotope signatures of nitrate and nitrate concentrations throughout the observation period, the integrated analysis with a Rayleigh plot and an isotope cross-plot (Fig. 7) clearly suggests a minor impact of

denitrification at the catchment scale. Fitting the Rayleigh equation to the observed data only yields an apparent, field-based enrichment factor that is normally smaller than the intrinsic enrichment factor that would be observed under closed system conditions (e.g., Druhan and Maher, 2017; van Breukelen, 2007). Despite this uncertainty, the obtained value of -2.7 ‰ cannot be considered as indicative for straightforward denitrification (Knöller et al., 2011). Even though a minor number of samples undoubtedly show the impact of denitrification with elevated isotope values and low concentrations (Fig.

7a), the overall nitrate isotope-concentration pattern is controlled by dilution and other flow-related processes as well as by the isotopic nitrate source variability (Benettin et al., 2020; Lutz et al., 2020). Accordingly, the dual isotope plot (Fig. 7b) does not show a strong positive correlation between nitrogen and oxygen isotope signatures expressed by a so-called denitrification line with a slope between 0.5 and 1.



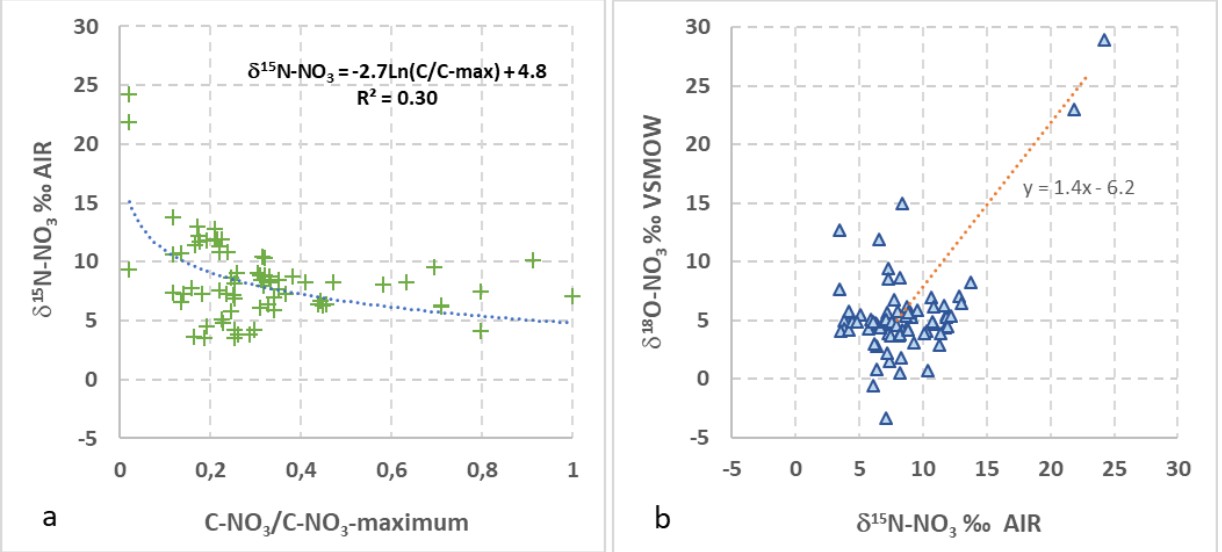

Figure 7: a. Relationship between nitrate concentrations (normalized to the highest measured concentration) and b. dual isotope plot showing the correlation between measured nitrogen and oxygen isotope signatures

## 3.2 Simulation results of the modified tran-SAS

### 3.2.1 Water isotopic signatures and water age distributions in discharge

In general, the simulated isotopic signature of stream water using the tran-SAS model mirrors the behavior of measured isotopic signature in stream water with a KGE of 0.48 (Fig. 8). Except for strong short-term fluctuations of the isotopic signatures in stream for instance in 2019 and at the end of 2020, the model is able to capture the general variation of isotopic signatures during individual years.

For the comparison of transit times of both water and nitrate (Fig. 10), the transit times of the routing storage are considered only. Transit times of water are lower during high flow conditions and higher during low flow conditions and long recession times. Discharge after winter and during spring of 2018, 2019 and 2020 becomes lower and by this the proportion of old water becomes higher, which is reflected by increasing median transit times. Water of different ages contributes to stream water with variable proportions over time. During the entire observation period between 2017 and 2020, the wettest conditions were observed in 2017. Correspondingly, the highest proportion of young water was found during a peak flow at the beginning of 2018, after the wet year 2017. The following years 2019 and 2020 were influenced by dry climate conditions with lowered discharge and increasing transit times of water. This implies that the higher proportion of older ages in stream flow during the dry year 2018 compared to the wet year 2017 is most likely related to higher relative contribution of older (ground-)water to the streamflow during the recession period and less young water from recent precipitation events, since less precipitation felt during the dry year 2018. The following years 2019 and 2020 are affected by drought conditions and overall less precipitation, causing higher old water contributions during summertime. It is most likely that the replenishment of water storages took a



few months after the drought in 2018. A similar behavior was reported by Smith et al. (2020) who analyzed the effect of the drought in 2018 on ecohydrological fluxes in Central Germany. According to their findings, the replenishment of the water storages took 6 to 8 months depending on the vegetation canopy. The impact of the drought can easily be followed by the discharge time series. Lowered discharge in subsequent years after 2018 reflect that the catchment has not yet recovered from the drought impact, as similarly found by Kleine et al. (2020).

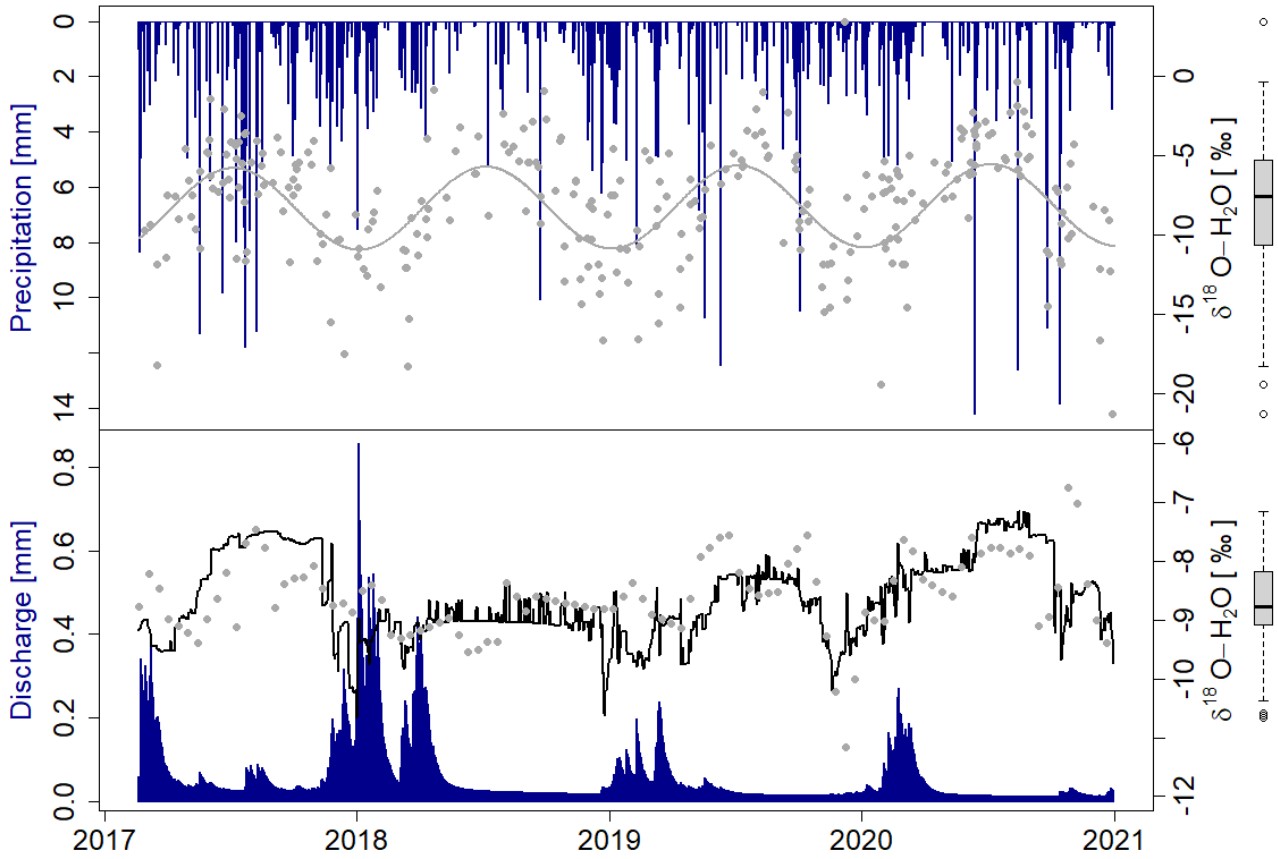


*Figure 8: Precipitation [mm] (blue) and isotopic signature of δ¹⁸O-H₂O in precipitation (grey: dots = measured, line = sinusoidal fitting line to precipitation isotopic signature) on top and discharge [mm] boxplots on the right side of the graph show the distribution of isotopic signature in precipitation (top) and streamflow (bottom)*

### 3.2.2 Nitrate isotopic signatures and TTDs gained from nitrate isotope model

The measured nitrate oxygen isotopic signature $\delta^{18}O\text{-}NO_3$ in stream water samples shows a considerable scatter (Fig. 4 and Fig. 9). Considering a range of literature values for isotope fractionation parameters during denitrification and degradation rates as proposed by Granger & Wankel (2016), we obtain a simulated scattering of oxygen isotopic signatures that is shown in orange in Fig. 9. Most of the measured values plot within that simulated range. Instead of applying variable fractionation





parameters, we used a simple model approach with constant degradation rates and constant fractionation factors during the
time series which is able to reflect the range of measured isotopic signatures. It is obvious though that biogeochemical
processes in soil and stream water are more complex and that $\delta^{18}$O-NO$_3$ is not only determined by nitrification and
denitrification. A significant share of nitrate with an oxygen isotopic signature not related to nitrification in any way can be
introduced into the catchment by direct input through inorganic fertilization using nitrate salts. That share can account for up
to 20-30% of the total fertilization in arable land (Boshers et al., 2019). Moreover, as already discussed in section 3.1 based
on the evaluation of the observed data, intermediates of redox reactions in the unsaturated and saturated zones involving
nitrogen transformations such as hydroxylamine or nitrite have the capability of quickly exchanging oxygen isotopes with the
ambient water and thereby introducing an uncertainty on the prediction of $\delta^{18}$O-NO$_3$ formed during nitrification according to
Eq. (8) (Buchwald and Casciotti, 2010; Casciotti et al., 2011; Boshers et al., 2019). Kool et al. (2011) state that the isotopic
signature $\delta^{18}$O-NO$_3$ may be completely controlled by $\delta^{18}$O-H$_2$O, due to the above-mentioned isotope exchange during
biogeochemical processes in the reactive zones. This potential exchange of $\delta^{18}$O and any water flow related mixing phenomena
of nitrate from different sources is not considered in our simple model, due to the fact that other mixing phenomena of nitrate
are not measurable with the data provided. Moreover, the isotopic signature of surrounding soil air has an influence on the
nitrified nitrate. In our study, we assumed that the proportion of oxygen from water and soil air are the same in soils as observed
in laboratory cultures and that, therefore, the incorporation of oxygen from soil air or water during nitrification is not associated
with isotope fractionation (Kendall et al., 2007). Besides, we assume that the isotopic signature $\delta^{18}$O of water used by microbes
is the same as the isotopic signature of water in the soil storage and that the isotopic signature $\delta^{18}$O of soil air used by microbes
is the same as the atmospheric isotopic signature (Kendall et al., 2007). However, under natural conditions in aquatic systems,
other processes can influence the $\delta^{18}$O of dissolved oxygen of soil air, e.g., the diffusion of atmospheric oxygen of air in the
subsurface as well as photosynthesis which lowers the $\delta^{18}$O of soil air, and respiration by microbes which leads to higher $\delta^{18}$O
values of soil air due to isotopic fractionation (Kendall et al., 2007; Boshers et al., 2019). In general, observed $\delta^{18}$O of nitrate
can show an offset compared to $\delta^{18}$O of nitrate computed with the simple Eq. (8), due to the implemented oxygen isotopic
signature of soil air which undergoes respiration (Kendall, 1998; Kendall et al., 2007). Besides, in addition to the common
autotrophic nitrification pathway, nitrate formation can occur to some extent via heterotrophic aerobic ammonia oxidation
(Mayer et al., 2001). Moreover, the proportion of oxygen from surrounding water and soil air can change to a minor degree
during nitrification (Aravena et al., 1993; Kool et al., 2011). It is still unresolved how these different mechanisms affect the
isotopic signature of nitrate during nitrification reactions and to what extent they can therefore not be adequately mirrored with
simple equations such as Eq. (8) (Kendall et al., 2007). Instead the applied equation gives a possible range of nitrate isotopic
signatures that could occur under the previously mentioned restrictions.

During wet periods, younger nitrate is dominant, which was released from the upper subsurface storages. In the Meisdorfer
Sauerbach catchment, we predominantly found transit times of water up to 300 days with lowest transit times (50 days) during
high flows. Hence, water with such short transit times easily transports soluble nitrate directly to the stream (Fig. 10).
Considering such short contact times between water-born nitrate and biofilms on the mineral matrix hosting denitrifying





microbial communities, we conclude that the impact of denitrification during the transport process at the catchment scale is relatively low. This assumption is clearly supported by an integrated analysis of observed field data (Fig. 7). Therefore, a low

denitrification rate constant was chosen during the calibration process.

*Figure 9: Panel A with the simulated nitrate concentrations [mg/l] of the mHM model of discharge in dark green and as leaching flux in light green. The black line in panel A shows the nitrate concentrations obtained from the nitrate isotope model. Panel B shows the isotopic signature of $\delta^{18}O$-$NO_3$ in stream: green dots= measured with measuring error 0.8 ‰, black line=*

*simulated, orange area= 10% best simulation according to a small bias between observed and simulated nitrate $\delta^{18}O$-$NO_3$ in stream.*

Once mobilized, the transport of nitrate within the catchment is expected to be closely linked to the transport pathways of water (Maher, 2010; Maher, 2011). Therefore, the TTDs of nitrate should display a similar behavior as the TTDs of water.





Considering the median transit times in the lower storage ($TT_{50}$) as shown in Fig. 10, nitrate has lower $TT_{50}$s than water

throughout the entire observation period, but both lines reflect the same seasonal behavior. The offset shown is caused by denitrification in the lower storage, as nitrate associated with short transit times contributes significantly more to the overall nitrate transit time because of its higher concentration compared to nitrate associated with large water transit times, which has been mostly denitrified before reaching the catchment outlet.

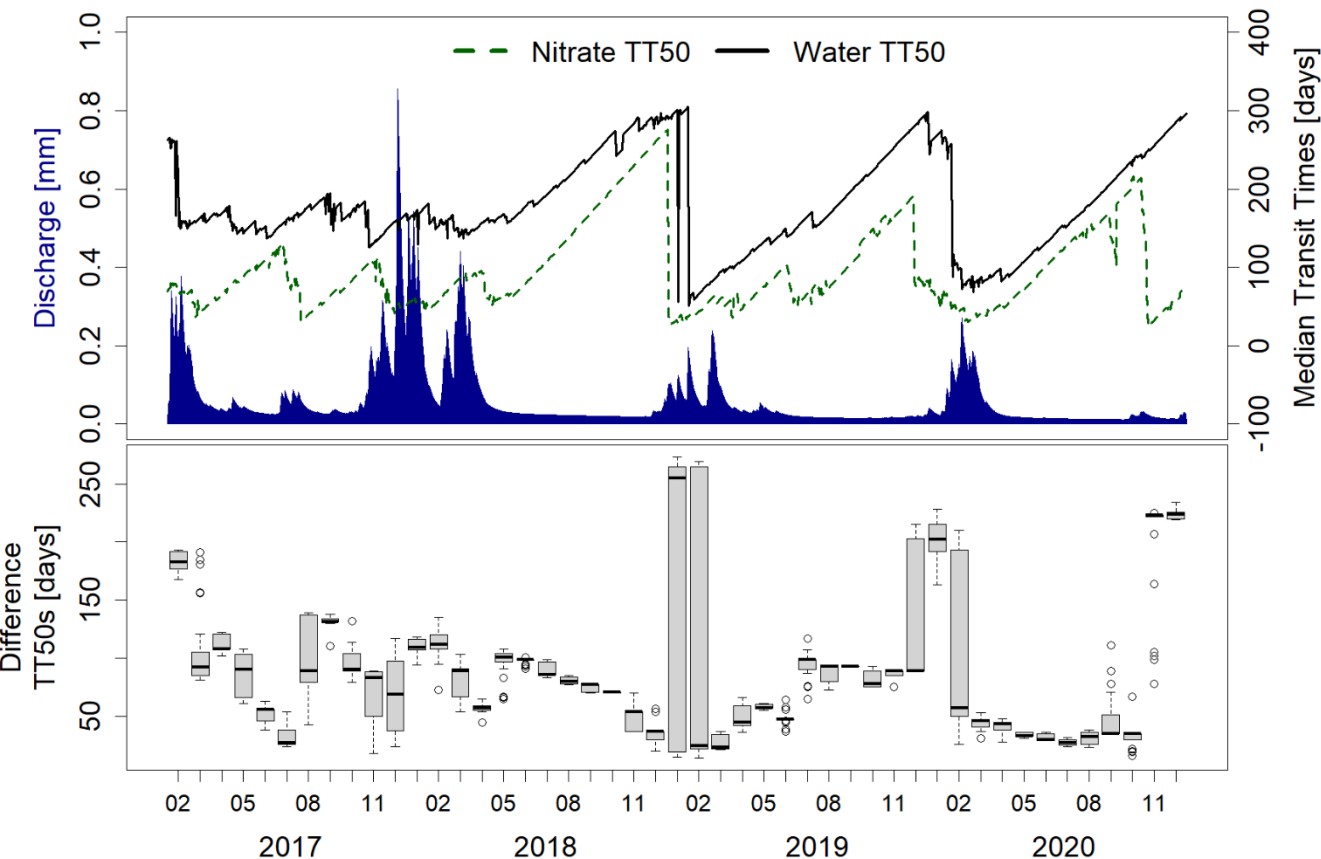

*Figure 10: Top panel: Median Transit Times ($TT_{50}$) of water (black solid line) and nitrate (green dashed line) through the lower storage, and discharge bottom panel: monthly boxplots of difference between $TT_{50 \, of}$ water and of nitrate*

Nevertheless, the offset between $TT_{50}$ of water and $TT_{50}$ of nitrate is not constant but shows a significant temporal variability. While the differences between $TT_{50}$ of water and $TT_{50}$ of nitrate are highest during periods with high discharge during winter time, lowest differences are observed during dry periods such as summertime (Fig. 10). The offset between $TT_{50}$ of water and

$TT_{50}$ of nitrate at the beginning of periods with high discharge (i.e. earlier lower $TT_{50}$ of nitrate than $TT_{50}$ of water in December 2018 and December 2019) is most likely caused by more old water contributions from deeper water sources such as groundwater that are active at the beginning of a high flow event before the young water from precipitation is dominating the





runoff process. The first flush after drier periods carries larger amounts of young nitrate through soil layers with fast flow paths, while older water contributing to runoff carries less amount of nitrate. Water transit times are known to decrease during

increasing discharge (Benettin et al., 2015; Soulsby et al., 2015).

Considering the low offset between both transit times during summertime in 2018, 2019 and 2020, one can assume that denitrification is lowered during these times. Even though high temperature is occurring during summer, the wetness of soils is lowered because of the prolonged drought conditions since 2018 (Kleine et al., 2020). During the year 2017 much more variability of nitrate transit times can be seen (Fig. 10). With our isotope data, we were not able to demonstrate significant

denitrification due to missing data. Nevertheless, the nitrate concentrations indicated that denitrification happened to a limited extent (Fig. 7). This is complemented by the relation between the transit time and nitrate concentration (Fig. 11): The lowered nitrate concentrations with higher transit times are an indicator for processes that degrade the solute nitrate along its transport path through the catchment.

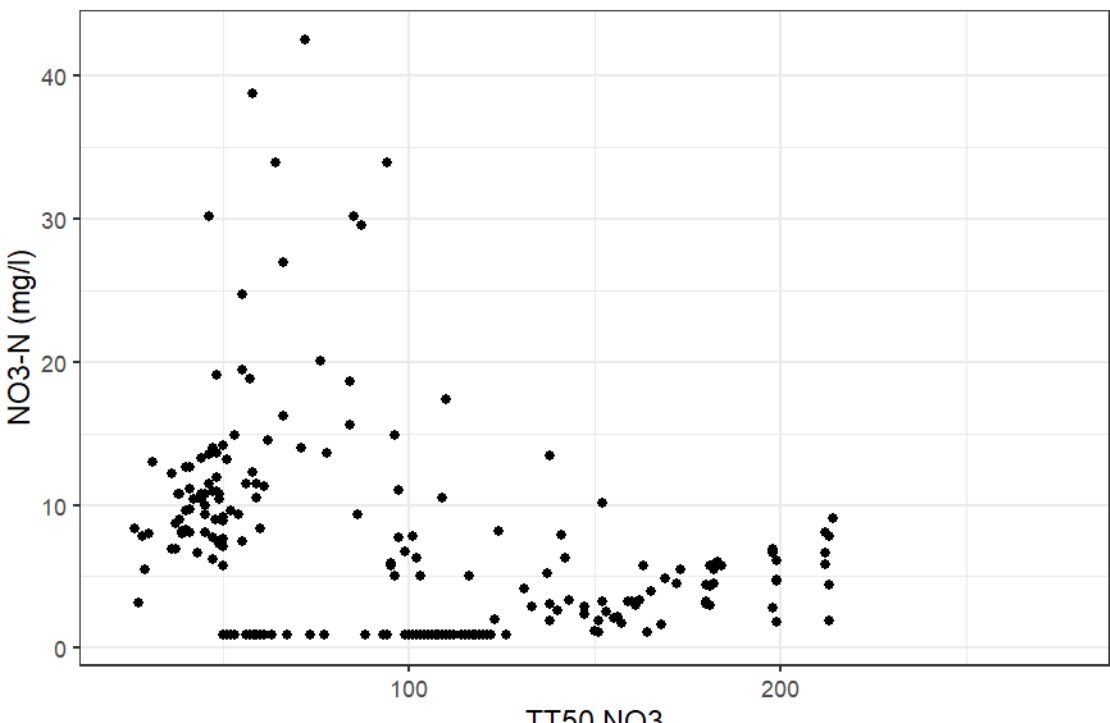

Figure 11: Relation between the measured nitrate concentration in stream water (NO$_3$-N [mg/l]) and the simulated median transit time of nitrate (TT$_{50}$ NO$_3$)

Considering the underlying geology with low to moderate groundwater connectivity, significant nitrate contributions to the stream from deeper aquifers hosting older groundwater do not seem very likely in the study catchment. Our results are in line with the findings of Jasechko et al. (2016) who point out that soluble contaminant inputs can be transmitted from watersheds





to streams during short time scales, which are in the range of young water fractions (~2 months), and especially in flat agriculturally dominated catchments where a higher proportion of young water is present, a faster release of solutes is possible. Especially for farmers it is relevant to improve the understanding of the processes in the catchments with the aim to prevent ecological habitats such as water bodies from high nitrate loads that cause eutrophication and are in general harmful to the ecosystem.

In general, the computation of water and nitrate transit times does not only have scientific, but also practical implications, as a relevant goal for farmers is to increase the efficiency of fertilizer application on agricultural fields so that nutrients are available for the crops as long as possible without being flushed out during precipitation events. Knowing how fast nitrate is released to the stream, farmers can improve their management practices, e.g., by reducing the amount of nitrate fertilizers that are applied at several times per year, but also by considering that even when it is allowed to apply fertilizer (e.g., until the end

of October in Germany), the weather forecast has to be considered more strongly with the aim to avoid fertilizer application during precipitation intense periods.

## 4 Potential impact of conceptual simplifications on the model performance

The deliberately chosen parsimonious modelling approach provided valuable insights into the transit time dynamics of water and nitrate in the investigated catchment. Nevertheless, we are aware that this relatively simple concept might be associated

with higher uncertainties with respect to the model output.

One issue that might have some impact on the model performance is the data collection strategy providing the data sets implemented in our model. For technical reasons, stream water samples were collected as grab samples at specific times of the day. Birkel et al. (2010) analyzed the effect of different sampling strategies such as composite samples versus grab samples and how model performances of different model types are affected by samples taken in different timesteps. They report a

decreasing model efficiency related to lowered sampling frequencies. Moreover, especially for catchments that release water predominantly from shallow subsurface storages, the influence of composite samples in lowered sampling frequencies causes less accurate model performances (Birkel et al., 2010). The fact that the Meisdorfer Sauerbach catchment releases predominantly water from the shallow subsurface during the investigated time series as well as the low sampling frequency during 2013 to 2017 may be responsible for the relatively low Kling-Gupta-Efficiency of 0.48 for stream water isotopic

signatures.

Regarding the modifications made for the nitrate isotopic signatures in tran-SAS, our model concept was deliberately simple. The calculation of the oxygen isotopic signature of nitrate generated during nitrification is based on a straightforward implementation of $\delta^{18}O$ of water and $\delta^{18}O$ of soil air according to Eq. (8) with a fixed isotopic signature of soil air to 23.5 ‰. Even though it is well known that the isotopic signature of soil air can vary depending on the biogeochemical processes that

occurred in the soil zone, other studies already evaluated that an isotopic signature of 23.5 ‰ is able to represent the overall isotopic signature in soil storages. For calculating $\delta^{18}O\text{-}NO_3$, we applied the simplified assumption that water oxygen is incorporated from the leaching flux leaving the upper storage. Compared to the $\delta^{18}O\text{-}H_2O$ of the precipitation input function,





the leaching flux shows a damped signal with a lower variability of the simulated $\delta^{18}$O-H$_2$O values. Under natural conditions, however, some of the nitrate will be formed with water that has not reached the same level of isotope signal dampening as

expected for the leaching flux. As a consequence, the overall variability of the observed $\delta^{18}$O-NO$_3$ could be slightly higher compared to the computed $\delta^{18}$O-NO$_3$.

By assuming that denitrification is the only process affecting the isotopic signatures of nitrate after initial nitrification, we applied a further simplification for our model concept.

Our main goal was to describe the age of nitrate compared to the age of water using median transit times as age metric gained

from a high-frequency isotope dataset. We assume, it is most important to consider, that nitrate is transformed during the transport with water, whereas water does not have a transformation. Therefore, we state that with the consideration of the denitrification of nitrate on its way through the catchment, our simple model gives a very relevant insight to the decoupling of nitrate and water transit times. Despite the simplifications and related uncertainties described above, our approach provides a novel tool that is fully in line with our intention to provide information on the differences or similarities of the age metrics of

water and nitrate in a mixed land-use headwater catchment.

**5 Conclusion**

Based on time series observed between 2017 and 2020, we investigated the transit time distributions of water and nitrate using their oxygen isotopic signatures ($\delta^{18}$O) as a characteristic fingerprint in concert with a simple model. The numerical model tran-SAS (Benettin and Bertuzzo, 2018) was modified by introducing a second storage and by applying simple biogeochemical

equations to describe nitrification and denitrification as well as associated isotopic signatures and isotope fractionations. The study was conducted in a 11.5 km² headwater catchment in the Northern lowland of the Harz mountains, Central Germany.

Generally, we found that nitrate transit times behave in the same way as water transit times, but with an apparent offset between median transit times of water and nitrate, which was highest at the beginning of high discharge periods due to higher contributions of water from water storages that contain old water such as groundwater storages with less nitrate. Instead, during

recession periods, predominantly young nitrate was released to the stream. Due to biogeochemical processes such as denitrification, the apparent transit time of nitrate can be lower than that of water, because the old nitrate has been degraded and, therefore, contributes less to the overall nitrate transit time. Hence, predominantly young nitrate is released to the stream. This information is highly relevant for understanding processes that control nitrate export from the agricultural fields to surface water ecosystems that are stressed by the impact of high nitrate loads. Moreover, this knowledge may be used to enhance

farming practices with the aim to increase the efficiency of fertilizer application on agricultural fields. For instance, to be sure that the loss of nutrients from fertilizer application is lowered, a buffer time before and after wet periods such as high precipitation events that cause the discharge to rise, could be considered for the planning of fertilizer application.

We conclude that our findings are characteristic for a mixed land use headwater catchment under significant hydrological and ecological stress associated with increasing drought conditions due to climate change. However, the assumptions we made

cannot necessarily be transferred to other catchments displaying largely different hydro-meteorological, topographic and/or



land use boundary conditions. Therefore, a broader investigation involving catchments of various characteristics is advisable in order to provide a more general view on the link between such catchment characteristics and transit times of water and nitrate.

Generally, our findings regarding the varying offset between water and nitrate transit times underline the importance of 545 analyses of solute transport and transformation in the light of projected more frequent hydrological extremes (droughts and floods) under future climate conditions.

**Data and code availability**

The nitrate isotope model set up is described in the present publication, by using tran-SAS v1.0 (Benettin and Bertuzzo, 2018) 550 as a basis. Input data for the model simulations as well as field data and mHM-Nitrate (Yang, et al., 2018; Yang & Rode, 2020) simulation data can be found at HydroShare: Radtke, C., K. Knöller, C. Müller, J. Rouhiainen (2024). Meisdorfer Sauerbach catchment measured data and model input data, HydroShare, http://www.hydroshare.org/resource/79f5635f893b4a90959b36ddb56aba8c.

**Author contribution**

CR and KK developed the study. Samples were taken by KK, CM, JR and CR. Laboratory analysis and the sample preparation have been conducted by CR, HW, JR and laboratory staff from the UFZ Halle. XY did the model simulation with the mHM-nitrate. CR developed with the support of SL and PB the nitrate isotope model, based on the tran-SAS model. KK, RM and SL acted as supervisor for the study. CR analysed the data and the results and wrote the first draft of the manuscript. KK, RM, 560 SL, PB and XY reviewed the manuscript.

**Declaration of competing interest**

The authors declare that they have no known competing financial interests or personal relationships that could have appeared to influence the work reported in this paper.

**Acknowledgments**

Funding for this study was provided by the Helmholtz Research Program. Laboratory analyses were conducted by laboratory staff of Helmholtz Centre for Environmental Research in Halle, Germany. Many thank to Tam V. Nyguen and Rohini Kumar for their support regarding the model set ups.

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
