# Peer review of "Nitrate and Water Isotopes as Tools to Resolve Nitrate Transit Times in a Mixed Land Use Catchment"

_Hydrology and Earth System Sciences, 2024_

## Referee Comment (RC2)

This is a review of "Nitrate and Water Isotopes as Tools to Resolve Nitrate Transit Times in a Mixed and Use Catchment" by Radtke et al.

The manuscripts proposes a model that combines isotopic signature of $^{18}O$ in stream water in both water and nitrate to understand the relation between water and nitrate transit times and the impact of denitrification processes.

The approach is innovative and original and could represent a significant advancement in the use of transit time distributions and isotope as tracers. The contribution can thus potentially be of interest for HESS and its readership. There are however some unclear parts and missing details that prevent me to fully evaluate the content at this stage. They are detailed below.

**Model formulation**

I appreciate that the authors decided to summarize the SAS model here in order for the paper to be self-contained, however, few more lines are needed to define variables and explain all the model details. Please add the definitions of the age ranked storage, the normalized age-ranked storage, and of the age "T". I guess parameters KQ1 and KQ2 (and KL1 and KL2) are the parameters of a (linear ?) relation between $k$ and the wetness index, but which one is which?

The model mentions evapotranspiration fluxes from the upper and lower storages, but I do not see how fractionation due to evaporation is addressed. I assume the authors assumed that evaporation is negligible with respect to transpiration. If so, please say this explicitly. (I also note that line 179 mentions only evaporation). However, on average stream water seems heavier than precipitation (Figure 8) in the data and in the model as well, pointing to some evaporation. Was this addressed somehow in the model?

If I understand correctly line 205, discharge used in the SAS calculation is the output of the mHM-Nitrate model, not the measured one. This choice has its benefit as it is important to have consistent leaching and discharge fluxes. However a reader is curious to see how well the model is reproducing the observed discharge to fully evaluate the results. Please add a comparison between the two (for instance in Figure 3).

Equations 2-5 use the symbol T for the age (although not defined), while equation 9 uses $\tau$. Is this used to make a distinction between age and transit time? Most papers on the topic uses the same symbol, indeed they are both ages evaluated for different samples. I think using two different symbols could be confusing.

Overall, I suggest the author to review with care the symbology used. For instance symbol $k$ is used for both the SAS parameter and the denitrification rate. "1" and "2" refer to the two parameter relation for $kQ$ and $KL$, but to the storages for parameter $K\_ET$. Also the use of superscripts and subscripts is not consistent across the manuscript and the symbol $\delta$ is not rendered in the pdf of the manuscript in a couple of places. Having a clear, intuitive and consistent symbology is crucial for the readability of the manuscript.

**Parameter estimation**

I do not follow how parameter estimation was performed. The model can produce as output $C_{NO_3}(t)$, $\delta^{18}O-NO_3$, and $\delta^{18}O-H_2O$, and all three variables can be compared with the observations in order to estimate parameters. However, from line 262 it seems that only the isotopic signatures were used. Why was nitrate concentration excluded? It does contain crucial information to constrain the denitrification rate.

Moreover, how was the metrics of KGE of water and nitrate $\delta^{18}$ combined to select parameters? It is not possible to consider $\delta^{18}O-NO_3$ and $\delta^{18}O-H_2O$ together (i.e. compute a unique KGE) because they have a completely different mean and standard deviation.

Line 267, caption of table 1 and line 370 mention the KGE of only water and not nitrate, which is confusing. How was the nitrate isotopic signature used? Finally, if the 10% best KGE simulations of 10000 parameter sets was retained, why table 1 reports 212 simulations instead of 1000? Overall, the parameter estimation procedure needs much more details to be understood and reproduced.

Please in Table 1 report also the lower and upper bounds of the range from which parameters were randomly extracted. Also please report, maybe as supplementary material, the classical GLUE plot with the scatter of each parameter value against the performance metric used, highlighting the behavioral sets.

**Model results**

Figure 8. Please report the range of the behavioral simulations in the modeled isotopic signature of water in discharge.

I do not follow lines 396-398 as they seem to contradict the caption of figure 9. In the caption the orange range is described as the 10% best simulations, while in the text the authors refer to a range of possible parameters taken form the literature, and not the results of the GLUE approach. Please clarify (this is related to the confusion about the performance metric used in the GLUE). Also the statement "Most of the measured values plot within that simulated range" is clearly not in agreement with the results reported in Figure 9 where just few points are within the orange range.

Line 440. I do not understand the sentence: "Orange area= 10% best simulation according to a small bias between observed and simulated nitrate δ18O-NO3 in stream." What do the authors mean by "according to a small bias"?

Figure 9A. I am curious about the difference between the nitrate concentration simulated by the two models. Does the mHM model include a more complex set of processes? But most importantly, **how do the two models compare with the data?** The comparison must be reported to interpret and fully evaluate model results. Maybe the authors can add a different panel with the comparison because the concentration in the leaching is off scale for a visual comparison.

Figure 10B. Please report also the isotopic signature in the leaching flux. I think it is crucial to understand the dampening and fractionation that occur in the routing storage.

Line 435. This seems to indicate that the denitrification rate and fractionation factor were assumed, not calibrated, as Table 1 suggests. The treatment of the nitrate model parameters need to be clarify because it is very unclear at the moment.

Figure 10. Please report also the variability of $TT_{50}$ related to the different behavioral parameter sets .

To summarize, it is unclear how nitrate data (concentration and isotopic signature) were used in parameter estimation. At points it seems that they were not used at all, which would represent a weak point of the manuscript that, in my view, must be revised for further consideration.

**Nitrate Transit Time**

The focus variable of the manuscript (starting from the title) is the nitrate transit time. However, details about its calculation are not provided. From the definition "time from its formation during nitrification in the soil until nitrate release to the stream" (line 50), I would proceed to the calculation of the transit time distribution of nitrate in the discharge ($p_{NO_3,Q}$) as follows:

$$p_{NO_3,Q}(\tau,t) = \frac{\left[C_{S,^{18}O}(t-\tau)e^{-k_{18_O}\tau} + C_{S,^{16}O}(t-\tau)e^{-k_{16_O}\tau}\right]p_Q(\tau,t)}{C_{NO_3}(t)}$$

It this correct? I think it is not such a straightforward detail that can be omitted from the text. Also it is important to add the detailed calculation for the results to be reproducible.

Moreover, the analytical expression of $p_{NO_3,Q}(\tau,t)$ helps in understanding the factors controlling the difference between $p_{NO_3,Q}(\tau,t)$ and $p_Q(\tau,t)$. As the isotopic ratio of oxygen in nitrate in the soil is around $2*10^{-3}$ , for the purpose of calculating $p_{NO_3,Q}(\tau,t)$ , $C_{S,^{18}O}$ is negligible with respect to $C_{S,^{16}O}$, and therefore the above equation can be approximated as:

$$p_{NO_3,Q}(\tau,t) \approx \frac{C_{S,^{16}O}(t-\tau)e^{-k_{16_O}\tau}p_Q(\tau,t)}{C_{NO_3}(t)}$$

It is interesting to note that if $C_{S,^{16}O}$ is fairly constant in time, or fluctuates around a mean value, the difference between nitrate and water transit time is driven by the denitrification rate $k$, and mathematically, the median nitrate transit time is lower than the water trave time for any $k > 0$, and the difference increases with $k$. However, also in the absence of denitrification $k = 0$, differences between nitrate and water transit time could arise because of temporal fluctuations of $C_{S,^{16}O}$ . I think this preliminary considerations are useful to guide the reader in interpreting the results.

Line 442: "*Once mobilized, the transport of nitrate within the catchment is expected to be closely linked to the transport pathways of water (Maher, 2010; Maher, 2011). Therefore, the TTDs of nitrate should display a similar behavior as the TTDs of water.*" Please note that the expectation in the second sentence is incorrect, even for a passive solute. As highlighted in the equation above, even for $k = 0$ (passive solute) difference between TTDs of water and TTDs of solutes arise because of

temporal variation in the solute input concentration. Think at this simple counterexample: a system with continuous input of water but just an impulse of solute at time $t_0$. At any time $t>t_0$, the TTD of water is potentially greater than 0 for any TT, but only solute with TT=t-t0 can be found in the sample.

This set up a false premise for the following discussion (line 444-455) where differences between TT50 of water and nitrate are discussed only in terms of denitrification, while the seasonal fluctuations of the nitrate concentration in the leakage (Figure 9A) can partially explain such differences.

Line 456: "most likely caused" sounds strange as the authors are commenting model results, not observations, so they can reconstruct exactly what is driving the observed pattern.

Line 461-462. Similarly to the comment before, the offset is produced by a model that has a constant denitrification rate, how can thus such difference be assumed to be caused by lower denitrification rates?

Line 480-486. Shouldn't farmers be interested in the nitrate transit time in the upper soil layer, rather than in the lower soil layer. Also, the farmer perspective should employ a forward transit time approach, not a backward one like in this case.

Overall, in my view, the authors fail to make a strong case for the use of nitrate transit time as a useful metric. As discussed above, nitrate TTD depends on nitrification (leaching concentration), denitrification in the deep layer and water  TTD. I think that classical mass balance metrics about these fluxes are more informative than the summary metric of the TT50 of nitrate (or its difference with TT50 of water), which blends together information about the underlying processes and provides overall less information.

**Minor comments**

Line 382: "felt" should read "fell"

---

## Author Comment (AC2)

This is a review of "Nitrate and Water Isotopes as Tools to Resolve Nitrate Transit Times in a Mixed and Use Catchment" by Radtke et al.

The manuscripts proposes a model that combines isotopic signature of 18O in stream water in both water and nitrate to understand the relation between water and nitrate transit times and the impact of denitrification processes.

The approach is innovative and original and could represent a significant advancement in the use of transit time distributions and isotope as tracers. The contribution can thus potentially be of interest for HESS and its readership. There are however some unclear parts and missing details that prevent me to fully evaluate the content at this stage. They are detailed below.

> *Answer: Dear Reviewer, thank you for your valuable feedback. We appreciate that you acknowledge the relevance of our study. In the following, we want to answer your comments.*

**Model formulation**

I appreciate that the authors decided to summarize the SAS model here in order for the paper to be self-contained, however, few more lines are needed to define variables and explain all the model details. Please add the definitions of the age ranked storage, the normalized age-ranked storage, and of the age "T". I guess parameters KQ1 and KQ2 (and KL1 and KL2) are the parameters of a (linear ?) relation between $k$ and the wetness index, but which one is which?

> *Answer: The parameters are defined in lines 179 – 184. KQ are the parameters of the beta shape storage selection function for the outflow of the system and KL are the parameters of the beta shape storage selection function for the leaching from the upper storage to the lower storage. Depending on the wetness index the parameter changes, details can be found in the original publication of tran-SAS. In our manuscript, we don't want to present a fully new model, we want to focus on the measured data that we obtained and a tool (the extended model) that we used to analyse our data in more detail. The focus of this manuscript is not the presentation of the model development, instead we use the model as a tool for further analysis.*

The model mentions evapotranspiration fluxes from the upper and lower storages, but I do not see how fractionation due to evaporation is addressed. I assume the authors assumed that evaporation is negligible with respect to transpiration. If so, please say this explicitly. (I also note that line 179 mentions only evaporation). However, on average stream water seems heavier than precipitation (Figure 8) in the data and in the model as well, pointing to some evaporation. Was this addressed somehow in the model?

*Answer: To fully consider the evapotranspiration and the fractionation of isotopes, we would need measured isotopic signatures of evapotranspiration, which we don't have. Therefore, we decided to test, calibrate and validate the model on stream water isotopic signatures, wherein the evapotranspiration is indirectly considered. This is accomplished by calibration/validation using measured isotopic signatures in stream water that underwent evapotranspiration on its flow path through the catchment. Still, the fractionation of the isotopes during evapotranspiration is a process that we cannot consider as long as we don't have measured data that we can compare with the simulated data. We mentioned this from lines 177ff. We will correct evaporation to evapotranspiration in line 179.*

If I understand correctly line 205, discharge used in the SAS calculation is the output of the mHMNitrate model, not the measured one. This choice has its benefit as it is important to have consistent leaching and discharge fluxes. However a reader is curious to see how well the model is reproducing the observed discharge to fully evaluate the results. Please add a comparison between the two (for instance in Figure 3).

*Answer: The simulated discharge by mHM-nitrate has been calibrated against discharge measurements from the larger Selke catchment that includes the Meisdorfer Sauerbach sub-catchment, as mentioned in lines 210ff. The discharge measurement station at the Meisdorfer Sauerbach (shown in Fig. 1) had several issues that we would like to explain in the following. First, the measurement device is on the top of a pipe and estimates the water level from above. There is no sensor under water. Due to that, only the height of the water surface is measured. One of the biggest problems that occurred was sedimentation on the bottom of the pipe causing the water level to rise. During heavier rain events the pipe was flushed and a lot of sediment was on the bottom of the pipe afterwards. This scenario is very fluctuating and changes from day to day. One day the water transports most of the sediment out of the pipe, some days later, the more sediment settles in the pipe. Even though we maintained this station by ourselves and cleaned it as often as possible, we still had to cope with this issue. Due to that, the water level measurements and corresponding discharge calculations are affected by the sediment which causes higher estimates of discharge. Secondly, the measurement device had some technical errors and therefore we don't have a continuous measurement for the whole time span that we wanted to work on. Therefore, we decided to not use the measured discharge for any comparison, because we can't trust the data.*
*The figure below illustrates that on average the simulated discharge by the mHM-nitrate model matches the measured discharge at the Meisdorfer*

*Sauerbach. Please be aware that due to the sedimentation in the pipe, the black line of the measured data is not reliable.*

[Figure]

Equations 2-5 use the symbol T for the age (although not defined), while equation 9 uses $\tau$. Is this used to make a distinction between age and transit time? Most papers on the topic uses the same symbol, indeed they are both ages evaluated for different samples. I think using two different symbols could be confusing. Overall, I suggest the author to review with care the symbology used. For instance symbol $k$ is used for both the SAS parameter and the denitrification rate. "1" and "2" refer to the two parameter relation for $kQ$ and $KL$, but to the storages for parameter $K\_ET$. Also the use of superscripts and subscripts is not consistent across the manuscript and the symbol $\delta$ is not rendered in the pdf of the manuscript in a couple of places. Having a clear, intuitive and consistent symbology is crucial for the readability of the manuscript.

> *Answer: We appreciate your detailed feedback and we will go through it again and make it coherent.*

**Parameter estimation**

I do not follow how parameter estimation was performed. The model can produce as output $C_{NO3}(t)$, $\delta 18O - NO3$, and $\delta 18O - H2O$, and all three variables can be compared with the observations in order to estimate parameters. However, from line 262 it seems that only the isotopic signatures were used. Why was nitrate concentration excluded? It does contain crucial information to constrain the denitrification rate.

Moreover, how was the metrics of KGE of water and nitrate $\delta 18$ combined to select parameters? It is not possible to consider $\delta 18O - NO3$ and $\delta 18O - H2O$ together (i.e. compute a unique KGE) because they have a completely different mean and standard deviation. Line 267, caption of table 1 and line 370 mention the KGE of only water and not nitrate, which is confusing. How was the nitrate isotopic signature used?

> *Answer: The isotopic signature of discharge was used to calibrate and validate the hydrological fluxes in the model. Nitrate and its isotopes are transported by water through the catchment and therefore have the same transit time, if we consider theoretically nitrate as a conservative tracer. Therefore, we can consider the parameters that have been calibrated with the water isotopic signature in discharge to be the same for water and for nitrate transport on its flow path through the catchment. To also consider the degradation of nitrate, we also want to calibrate the degradation rate and for that we consider the isotopic signature of nitrate to estimate and calibrate the denitrification rate coefficient and the fractionation factor. With isotopic signatures of nitrate one can very precisely identify denitrification processes. A calibration with nitrate concentration is therefore negligible, because the calibration would be doubled, because the nitrate concentration is used to estimate the fractions of 16O-NO3 and 18O-NO3. To estimate the fraction of 16O-NO3 and the fraction of 18O-NO3 in the leaching nitrate we use the simulated nitrate concentration of the mHM-nitrate model. By using this concentration already in the computation of isotopic fractions, it would be indirectly influenced if we use the nitrate concentration for the calibration as well. Therefore, we decided to only calibrate the denitrification parameters on the isotopic signatures of nitrate in the stream water.*

Finally, if the 10% best KGE simulations of 10000 parameter sets was retained, why table 1 reports 212 simulations instead of 1000?

> *Answer: The 10% best KGE does not yield in 1000 out of 10.000, because we focus on the value of KGE and not on the amount of parameter sets by selecting the 10% best KGEs. We use the highest KGE e.g. 0.7 and want to have the 10% best according to the 10% closest KGEs. So the span is from all parameter sets with a KGE between 0.63 and 0.7.*

Overall, the parameter estimation procedure needs much more details to be understood and reproduced. Please in Table 1 report also the lower and upper bounds of the range from which parameters were randomly extracted. Also please report, maybe as supplementary material, the classical GLUE plot with the scatter of each parameter value against the performance metric used, highlighting the behavioral sets.

> *Answer: We reported the lower and upper bounds of the range of parameters in table S1 in the supplementary. Regarding the scatter of each parameter, we can acknowledge your suggestion, and we will add the following figure to the supplementary.*

[Figure]

**Model results**

Figure 8. Please report the range of the behavioral simulations in the modeled isotopic signature of water in discharge.

> *Answer: We will change that in the resubmitted manuscript and add the range of behavioural simulations.*

I do not follow lines 396-398 as they seem to contradict the caption of figure 9. In the caption the orange range is described as the 10% best simulations, while in the text the authors refer to a range of possible parameters taken form the literature, and not the results of the GLUE approach. Please clarify (this is related to the confusion about the performance metric used in the GLUE). Also the statement "Most of the measured values plot within that simulated range" is clearly not in agreement with the results reported in Figure 9 where just few points are within the orange range.

Line 440. I do not understand the sentence: "Orange area= 10% best simulation according to a small bias between observed and simulated nitrate δ18O-NO3 in stream." What do the authors mean by "according to a small bias"?

> *Answer: Thanks for pointing that out. We will define it more clearly in the manuscript. From the literature, we took the range of possible values for the fractionation factor and the denitrification factor. With the GLUE approach we tested 10.000 parameter sets with varying fractionation factors and varying denitrification factors. Only the 10% best value according to a high KGE value and a small bias between the observed and the simulated values are shown. We will correct the statement that most measured isotopic signatures plot within the simulation range to the statement that the scatter/temporal dynamics of the measured isotopic signatures of nitrate can not be easily mirrored by the model simulation, while the model simulation plots overall within the range of the measured isotopic signatures.*

Figure 9A. I am curious about the difference between the nitrate concentration simulated by the two models. Does the mHM model include a more complex set of processes? But most importantly, **how do the two models compare with the data?** The comparison must be reported to interpret and fully evaluate model results. Maybe the authors can add a different panel with the comparison because the concentration in the leaching is off scale for a visual comparison.

> *Answer: We will consider that in our resubmission of the manuscript. We will show a plot with measured data and the data obtained by the models.*

Figure 10B. Please report also the isotopic signature in the leaching flux. I think it is crucial to understand the dampening and fractionation that occur in the routing storage.

> *Answer: With the isotopic signature in the leaching flux one would only see the isotopic signature of nitrate from it's forming. The fractionation during the denitrification occurs when the isotopic signature of nitrate is released to the stream, where the transit time is considered. By that, the extend of denitrification is related to the time nitrate spent in the storage. Due to that, the isotopic signature of nitrate that underwent degradation is already visible in the stream.*

Line 435. This seems to indicate that the denitrification rate and fractionation factor were assumed, not calibrated, as Table 1 suggests. The treatment of the nitrate model parameters need to be clarify because it is very unclear at the moment.

> *Answer: As described, the range of possible values for the denitrification factor and fractionation factor have been taken from literature. The calibration yielded the best results with lower denitrification rates. It is not well written at this line and we will rewrite this part for resubmission.*

Figure 10. Please report also the variability of TT50 related to the different behavioral parameter sets .

> *Answer: We will add a plot with the minimum, median and maximum transit time of the 10% best parameter sets in the resubmitted manuscript.*

To summarize, it is unclear how nitrate data (concentration and isotopic signature) were used in parameter estimation. At points it seems that they were not used at all, which would represent a weak point of the manuscript that, in my view, must be revised for further consideration.

> *Answer: We described more clearly how isotopic signatures and nitrate concentration have been used for parameter estimation.*

**Nitrate Transit Time**

The focus variable of the manuscript (starting from the title) is the nitrate transit time. However, details about its calculation are not provided. From the definition "time from its formation during nitrification in the soil until nitrate release to the stream" (line 50),

I would proceed to the calculation of the transit time distribution of nitrate in the discharge ($pNO3,Q$) as follows:

$$p_{NO_3,Q}(\tau, t) = \frac{\left[C_{S,^{18}O}(t - \tau)e^{-k_{^{18}O}\tau} + C_{S,^{16}O}(t - \tau)e^{-k_{^{16}O}\tau}\right]p_Q(\tau, t)}{C_{NO_3}(t)}$$

It this correct? I think it is not such a straightforward detail that can be omitted from the text. Also it is important to add the detailed calculation for the results to be reproducible. Moreover, the analytical expression of $pNO3,Q(\tau, t)$ helps in understanding the factors controlling the difference between $pNO3,Q(\tau, t)$ and $pQ(\tau, t)$. As the isotopic ratio of oxygen in nitrate in the soil is around 2*10-3 , for the purpose of calculating $pNO3,Q(\tau, t)$ , $C_{S,O}18$ is negligible with respect to $C_{S,O}16$ , and therefore the above equation can be approximated as:

$$p_{NO_3,Q}(\tau, t) \approx \frac{C_{S,^{16}O}(t - \tau)e^{-k_{^{16}O}\tau}p_Q(\tau, t)}{C_{NO_3}(t)}$$

It is interesting to note that if $C_{S,O}16$ is fairly constant in time, or fluctuates around a mean value, the difference between nitrate and water transit time is driven by the denitrification rate $k$, and mathematically, the median nitrate transit time is lower than the water trave time for any $k > 0$, and the difference increases with $k$. However, also in the absence of denitrification $k = 0$, differences between nitrate and water transit time could arise because of temporal fluctuations of $C_{S,O}16$ . I think this preliminary considerations are useful to guide the reader in interpreting the results. Line 442: "*Once mobilized, the transport of nitrate within the catchment is expected to be closely linked to the transport pathways of water (Maher, 2010; Maher, 2011). Therefore, the TTDs of nitrate should display a similar behavior as the TTDs of water.*" Please note that the expectation in the second sentence is incorrect, even for a passive solute. As highlighted in the equation above, even for $k = 0$ (passive solute) difference between TTDs of water and TTDs of solutes arise because of temporal variation in the solute input concentration. Think at this simple counterexample: a system with continuous input of water but just an impulse of solute at time t0. At any time t>t0, the TTD of water is potentially greater than 0 for any TT, but only solute with TT=t-t0 can be found in the sample. This set up a false premise for the following discussion (line 444-455) where differences between TT50 of water and nitrate are discussed only in terms of denitrification, while the seasonal fluctuations of the nitrate concentration in the leakage (Figure 9A) can partially explain such differences.

*Answer: We agree that even a passive solute can only be found in the system, if it has been applied in the field. In the case of a controlled, closed system with fixed boundaries, we only can find a solute if it has been applied. Our system is a sub-catchment with an area about 11.5 km², while nearly half of the area is agricultural with the potential of very frequent fertilizer application. It is an open system with many different inputs, as well as diffuse sources and point sources. We never know exactly how much nitrate has been applied in the whole area and how often. We can assume that farmers respect the guidelines of fertilization. Besides, it is unknown how much nitrate flows into the sub-catchment from neighbouring and surrounding catchments. Furthermore, there will be an amount of nutrients that is stored in the ground for a longer time period because there are anoxic conditions that make it impossible for biogeochemical processes to occur. By that, it is also possible that there is nitrate in the catchment, that is much older than that which was formed after recent fertilizer application. The older nitrate can be remobilized after water flushes that part of catchment storage. So, in an open system like our sub-catchment we can assume that there is always a specific amount of nitrate, which is complemented by the continuous nitrate flux in the stream (Fig. 9). In the following figure, the measured nitrate concentrations can be compared with the simulated nitrate concentrations by the mHM-nitrate.*

[Figure]

*With this knowledge about the open system /catchment, we state that the transit time of water and the transit time of a degradable solute such as nitrate only differs due to the degradation of the solute. The oldest amount of nitrate underwent more denitrification than younger nitrate and therefore, there is less old nitrate and more young nitrate.*

*Considering the denitrification factor, there is a difference between the extent of denitrification of 18O-NO3 and 16O-NO3. Therefore, individual denitrification rates and fractionation factors are necessary.*
*The term that has been used to produce the age distribution of the individual nitrate isotopic signatures is mentioned in line 242, Equ. 9.*

Line 456: "most likely caused" sounds strange as the authors are commenting model results, not observations, so they can reconstruct exactly what is driving the observed pattern. Line 461-462. Similarly to the comment before, the offset is produced by a model that has a constant denitrification rate, how can thus such difference be assumed to be caused by lower denitrification rates?

> *Answer: Even though we built the model and want to describe the model results, we try to interpret these findings from the model simulation and we try to put them into a natural context. Therefore, we wrote "most likely", because there can be other interpretations as well.*

Line 480-486. Shouldn't farmers be interested in the nitrate transit time in the upper soil layer, rather than in the lower soil layer. Also, the farmer perspective should employ a forward transit time approach, not a backward one like in this case. Overall, in my view, the authors fail to make a strong case for the use of nitrate transit time as a useful metric. As discussed above, nitrate TTD depends on nitrification (leaching concentration), denitrification in the deep layer and water TTD. I think that classical mass balance metrics about these fluxes are more informative than the summary metric of the TT50 of nitrate (or its difference with TT50 of water), which blends together information about the underlying processes and provides overall less information.

> *Answer: There have been many interesting studies about nitrate concentrations in catchments that have their value. With isotopic signatures we can more precisely track processes that occurred in a system. The proposed nitrate TTDs make use of the measured isotope information, that can directly reflect internal nitrate transformations during its transport pathways. This is a unique advance compared to conventional flux-based metrics. Besides the commonly used flux based metrics, the nitrate TTDs is intended to provide as a complementary metric that can help better capture nitrate fates throughout the catchment system. Here, we show a first attempt how these isotopic signatures of nitrate can be used to get a better understanding of the biogeochemical processes in a catchment. We used the backward TT approach as a first attempt to reproduce what happened in the past. For farmers, it would be interesting to know more about a forecasting of*

*how long nitrate will stay in the system as well as for water quality management plans, but this would be a next step. This manuscript is aimed to improve the understanding of how long nitrate travels through the catchment and to make awareness that nitrate stays much longer in the system than it might be expected. We are not showing here the end of a chapter that has to be closed afterwards. Instead we want to show possible other tools to get more precise information about what is happening in a black box like catchments. It is a starting point and might be useful for further research.*

**Minor comments**

Line 382: "felt" should read "fell" – *Thanks, we will correct that.*